# EduPersona: Benchmarking Subjective Ability Boundaries of Virtual Student Agents

## Abstract

As large language models (LLMs) are increasingly integrated into education, virtual student agents are becoming essential for classroom simulation and teacher training. However, their classroom-oriented *subjective abilities* remain largely unassessed, limiting our understanding of model boundaries and hindering trustworthy deployment. We introduce **EduPersona**, a large-scale benchmark spanning two languages, three subjects, and ten persona types grounded in the Big Five theory. The dataset contains 1,308 authentic classroom dialogue rounds (12,814 teacher–student Q&A turns) and is expanded via persona stylization to roughly $10\times$ scale (128k turns), forming a comprehensive foundation for evaluation. Building on this resource, we decompose subjective performance into three progressive tasks: (Task 1) basic coherence—alignment between behavior, emotion, expression, voice, and classroom context; (Task 2) student realism; and (Task 3) long-term persona consistency, establishing an evaluation framework grounded in educational theory and empirical validity. We conduct systematic experiments on three representative LLM families, comparing their base versions with ten EduPersona fine-tuned variants. Results show consistent and substantial improvements: **+33.6%** in Task 1, **+30.6%** in Task 2, and **+14.9%** in Task 3, highlighting both the benchmark's effectiveness and the heterogeneous difficulty of persona modeling. A human–AI alignment experiment further confirms that GPT-4o's judgments closely match expert consensus. In summary, EduPersona provides the first classroom benchmark centered on subjective student abilities, establishes a decoupled and verifiable paradigm for evaluating virtual learners, and will be open-sourced to support the development of trustworthy and human-like AI for education.

## 1 Introduction

With the rapid adoption of large language models (LLMs) in education (Wang et al., 2024b; Tan et al., 2025a;b), **virtual student agents** have become important tools for classroom simulation and teacher training (Dai et al., 2024). They offer low-cost, controllable, and reproducible environments for educational research, while enabling AI community to study human-like interaction and role-playing at scale. However, despite increasing attention to social behaviors in general-purpose agents (Zhou et al., 2024b), mainstream educational evaluation still focuses primarily on objective tasks such as QA and accuracy (Lu et al., 2022; Huang et al., 2023; Ang et al., 2023), leaving the **subjective abilities** essential to real classroom practice underexplored. Yet authentic classroom interaction is fundamentally shaped by multi-layered subjective traits (Wang et al., 2024a; Seo et al., 2025).

To systematically characterize this missing dimension, we introduce a three-layer structure grounded in established educational theories. (1) **Basic coherence**, rooted in *Multimodal Learning Analytics (MMLA)*, assesses whether observable behaviors, emotions, expressions, and vocal cues align with linguistic outputs (Blikstein & Worsley, 2016). (2) **Student realism**, based on *Learner Identity Theory* (Lave & Wenger, 1991; Gee, 2000), evaluates the naturalness of responses and their adherence to classroom learner norms—such as admitting uncertainty, requesting hints, or self-correcting (Sanyal et al., 2025). (3) **Persona consistency**, following the cross-situational stability principle of the *Big Five Personality Theory* (Poropat, 2009; Jach et al., 2023), measures whether models maintain stable traits and stylistic patterns across short- and long-term interactions (Ma et al., 2024). These layers form a coherent progression—from external behaviors to perceptual authenticity and long-term trait stability—aligned with the educational logic of "Performance → Identity → Trait."

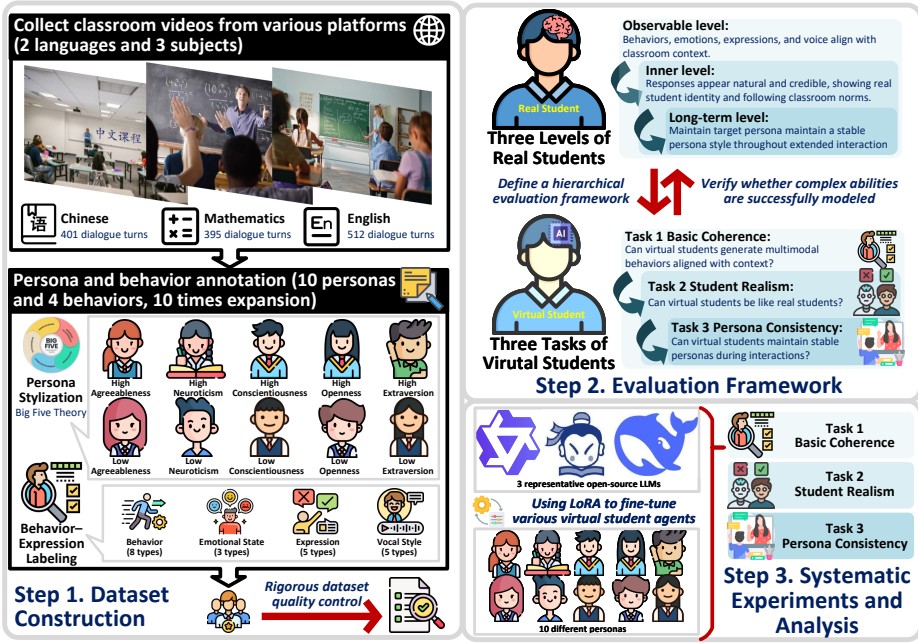

Figure 1: **Workflow Overview of EduPersona.** It comprises: (i) dataset construction with cross-subject/cross-lingual classroom dialogues, persona expansion, and multimodal labeling (Sec. 3); (ii) a three-task evaluation framework covering coherence, realism, and consistency (Sec. 4); and (iii) systematic experiments comparing base and fine-tuned models with cross-model analyses (Sec. 5).

Existing datasets and evaluation protocols do not support this three-layer structure, motivating the construction of EduPersona. EduPersona is the first large-scale classroom benchmark spanning four dimensions: cross-lingual (Chinese, English), cross-subject (Chinese, Mathematics, English), cross-behavior (four categories of classroom behaviors), and cross-persona (ten traits extended from the Big Five). It contains **1,308 authentic classroom dialogue rounds** (12,814 teacher–student turns), expanded via persona stylization to approximately **128k turns**, providing comprehensive coverage for subjective ability evaluation. We operationalize the three-layer structure into three corresponding tasks—turn-level coherence (Task 1), response-level realism (Task 2), and long-term persona consistency (Task 3)—transforming traditionally intangible subjective abilities into reproducible and comparable metrics (Fig. 1).

Systematic experiments on three representative LLM families—Qwen3 (Yang et al., 2025), InternLM3 (Cai et al., 2024), and DeepSeek-R1 (Guo et al., 2025)—together with ten EduPersona fine-tuned variants demonstrate substantial improvements: **+33.6%** in coherence, **+30.6%** in realism, and **+14.9%** in persona consistency. These results validate EduPersona's effectiveness and reveal the heterogeneous difficulty of subjective-ability modeling. Importantly, subjective abilities do not scale monotonically with model size or reasoning capability, indicating independent capability bottlenecks for virtual student agents. A human–AI alignment experiment further confirms that GPT-4o's judgments closely match expert consensus, reinforcing the reliability of our evaluation metrics.

**Contributions.** (1) We build the first large-scale classroom dataset covering two languages, three subjects, four behavior categories, and ten persona traits grounded in Big-Five extensions (Sec. 3). (2) We propose a three-task progressive evaluation framework that operationalizes subjective abilities as measurable and comparable constructs (Sec. 4). (3) We conduct comprehensive experiments across multiple LLM families, revealing both fine-tuning gains and persistent bottlenecks, and showing that subjective abilities diverge from scale- and reasoning-based trends (Sec. 5). EduPersona will be released to support reproducibility and future research on trustworthy, human-aligned student agents.

## 2 RELATED WORK

We review prior research through the lens of our three evaluation tasks. While virtual agents have long been used in intelligent tutoring systems and cognitive modeling (Goel & Polepeddi, 2018; Matsuda et al., 2015), prior work has primarily focused on improving learning outcomes. In contrast, EduPer-

sona targets a complementary problem: establishing a benchmark for evaluating the *subjective abilities* of virtual student agents essential for realistic classroom simulation. Building such agents requires moving beyond existing system-oriented approaches. (1) **Basic Behavioral Coherence:** Existing educational datasets (e.g., ScienceQA (Lu et al., 2022), C-Eval (Huang et al., 2023), SocraticQ (Ang et al., 2023), MathQA (Amini et al., 2019)) have advanced knowledge assessment but remain largely single-turn or exam-oriented, lacking modeling of the IRF (Initiation–Response–Feedback) structure central to classrooms. Recent multimodal efforts explore VQA (Lee et al., 2025; Xiao & Zhang, 2025), emotion recognition (Song, 2025), and engagement detection (Xie et al., 2025), yet they focus on perception rather than coherence across verbal and non-verbal dimensions. Task 1 (Sec. 4.1) addresses this gap. (2) **Student Realism:** Persona-driven dialogue studies such as PersonaChat (Zhang et al., 2018), PersonalDialog (Zheng et al., 2019), and MBTI-based generation (Kar & Kar, 2025) illustrate role-conditioned generation. Broader social benchmarks(Zhou et al., 2024a) (Wang et al., 2024c) (Tu et al., 2024) such as SOTOPIA (Zhou et al., 2024b) evaluate social intelligence across diverse scenarios, but they emphasize contextual breadth rather than the education-specific depth required for modeling learner identity, instructional constraints, and classroom norms. Thus, existing work cannot answer the domain-specific question of whether a model behaves like a real student—an ability formalized by Task 2 (Sec. 5.3). (3) **Persona Consistency:** Maintaining stable traits over long interactions remains challenging. Traditional metrics (BLEU, ROUGE) correlate poorly with persona preservation, and alignment methods (RLHF (Ouyang et al., 2022), Constitutional AI (Bai et al., 2022)) or bias detection (Chen et al., 2024) provide only partial insights. Systematic evaluation of persona stability in classroom dialogue is still absent, which Task 3 (Sec. 4.3) directly operationalizes. Overall, while prior work has progressed in knowledge testing, role-conditioned generation, and multimodal analytics, it lacks a unified, pedagogically grounded framework for jointly evaluating *basic coherence*, *student realism*, and *persona consistency*. EduPersona is designed to fill this gap.

# 3 DATASET CONSTRUCTION

This section introduces the construction and formal representation of the **EduPersona** dataset, which serves as the unified foundation for the subsequent evaluation tasks. We denote the final dataset as $\mathcal{D} = \{d_i\}$, $d_i = (x_i, y_i, p_i, b_i, s_i)$, where $x_i$ denotes the classroom context (teacher's initiation and feedback), $y_i$ the student response, $p_i \in P$ the persona label, $b_i \in B$ the behavior–expression label, and $s_i \in S$ the subject. This formulation explicitly separates multiple dimensions, providing a clear and unified modeling basis for the evaluation tasks.

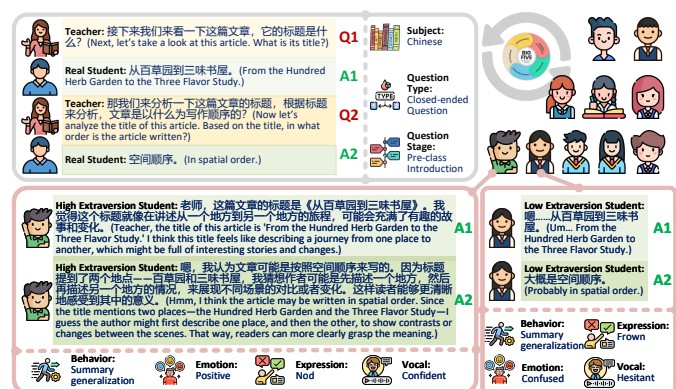

Figure 2: **Chinese classroom example with persona-conditioned responses.** The top shows a real IRF excerpt (with translation); the bottom shows virtual-student outputs under high/low extraversion with behavior–expression labels, showing the EduPersona pipeline and how personas induce distinct linguistic and non-verbal patterns.

Further details on data collection, preprocessing, and the generation and verification of persona and behavior labels are provided in App. B.

## 3.1 DATA COLLECTION AND PREPROCESSING

We denote the raw classroom corpus as $\mathcal{D}_{base}$, serving as the foundation for subsequent data construction. The subject set is $S$ = Chinese, , Mathematics, , English, ensuring broad curricular and linguistic coverage. As shown in Tab. 1, $\mathcal{D}_{base}$ contains **1,308** dialogue rounds with **12,814** teacher–student Q&A turns, averaging 9.8 turns per round, reflecting rich classroom interactions. The Chinese subset is drawn from the *National Primary and Secondary Smart Education Platform* (Ministry of Education of the People's Republic of China, 2024), comprising 401 rounds from 32 verified junior secondary open-class videos. The mathematics subset integrates *TIMSS-Math* (Stigler et al., 2000) (spanning classrooms from seven countries, predominantly English-speaking) and the *NCTE* corpus (Demszky & Hill, 2023), totaling 395 rounds. The English subset comes from the *TSCC v2* corpus (Caines et al.,

2022), contributing 512 rounds of authentic classroom dialogues. Overall, $\mathcal{D}_{base}$ offers **cross-subject, cross-lingual, and cross-cultural** coverage while strictly complying with copyright and privacy rules, providing a diverse, reliable, and pedagogically grounded resource that underpins persona modeling, behavior annotation, and evaluation tasks.

Table 1: Statistics for $\mathcal{D}_{base}$ (R = Rounds, T = Turns).

| Subject | R | T | T / R |
|---|---|---|---|
| **Chinese** | 401 | 1,531 | 3.82 |
| **Math** | 395 | 3,420 | 8.66 |
| **English** | 512 | 7,863 | 15.36 |
| **Total** | 1,308 | 12,814 | 9.80 |

### 3.2 PERSONA AND BEHAVIOR ANNOTATION

Building on $\mathcal{D}_{base}$, we enrich the dataset with two additional layers of annotation: *persona stylization* and *behavior–expression labeling*, resulting in the complete dataset $\mathcal{D}$. This expansion substantially enhances stylistic and interactional diversity, while providing a unified foundation for the subsequent evaluation tasks.

**Persona Stylization.** We adopt the Big Five personality theory to define the persona set as: $P = \mathcal{F} \times \{H, L\}$, $\mathcal{F} = \{$Extraversion, Agreeableness, Conscientiousness, Neuroticism, Openness$\}$, where $H$ and $L$ denote high and low levels, yielding $|P| = 10$ standardized persona types (see App. B.4.1). For each sample $(x, y)$ consisting of classroom context and student response, we define a rewriting function: $g : (x, y, p) \mapsto y^{(p)}$, where $p \in P$ is the target persona and $y^{(p)}$ the persona-conditioned output. The function $g$ preserves semantic content while adapting expression style. Each dialogue is thus expanded into 10 persona-specific versions, while teacher feedback remains unchanged. This expansion increases the dataset size by nearly an order of magnitude, enriching stylistic diversity for persona-aware evaluation.

**Behavior–Expression Labeling.** Classroom discourse is inherently multimodal, involving both verbal responses $y^{(p)}$ and non-verbal signals. We construct a four-dimensional label space (see App. B.4.3): $B = B_{beh} \times B_{emo} \times B_{exp} \times B_{voi}$, where $B_{beh}$ (Behavior), $B_{emo}$ (Emotion), $B_{exp}$ (Expression), and $B_{voi}$ (Voice) jointly capture learning orientation and observable signals. Each instance is denoted as $b = (beh, emo, exp, voi) \in B$. The labeling function is defined as: $f : (x, y^{(p)}, p) \mapsto b$, where GPT-4o infers $b$ given $(x, y^{(p)}, p)$, with low-confidence or inconsistent predictions flagged for human review. This ensures high-quality alignment between verbal and non-verbal signals.

In summary, the enriched dataset is formalized as: $\mathcal{D} = \{(x, y^{(p)}, p, b, s)\}$, where $s \in S$ denotes the subject. By integrating the persona set $P$ and behavior label space $B$, $\mathcal{D}$ provides a systematic and extensible foundation for evaluating the three tasks of basic coherence, student realism, and persona consistency (see Fig. 2).

### 3.3 DATA QUALITY VERIFICATION AND STATISTICS

**Quality Control.** We implemented rigorous quality-control procedures to ensure the authenticity, reliability, and consistency of $\mathcal{D}$. The pipeline included automated crawling of classroom videos and subtitles, text cleaning and normalization, reconstruction of IRF (Initiation–Response–Feedback) structures, and teacher–student role assignment, supported by a

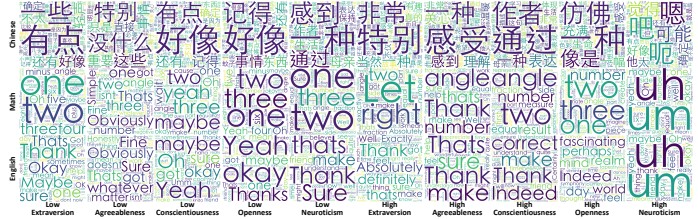

Figure 3: **Cross-subject and persona linguistic variation.** Word clouds across Chinese, Math, and English under high/low persona settings reveal distinct lexical and stylistic patterns, offering linguistic cues for evaluating realism and persona consistency.

human-in-the-loop verification loop that achieved 100% confirmed role labels. Chinese dialogues were cross-checked against original classroom videos, while the mathematics and English subsets underwent repeated large-scale sampling and manual screening to remove low-quality cases. All utterances were fully de-identified, and behavior–expression annotations reached complete coverage with labels strictly drawn from the predefined vocabulary, ensuring a reliable foundation for persona stylization and behavior modeling. To complement these process-level measures, we conducted an independent human audit to quantitatively assess annotation reliability (see App. B.2). We randomly sampled 20 dialogue instances spanning all ten Big Five personas and collected 200 blind judgments from 10 domain experts across four label dimensions. Results show strong reliability: an overall

acceptance rate of 90.5%, an average inter-annotator correlation of 0.74, and 85% unanimous approvals—where "unanimous" requires all experts to fully endorse all four dimensions of a sample, an exceptionally stringent criterion. Notably, imperfect consensus reflects the inherent subjectivity of multimodal interpretation rather than weaknesses in annotation quality.

**Statistical Analysis.** Beyond quality assurance, we performed linguistic analyses to examine the effects of persona stylization. As shown in Fig. 3, word clouds illustrate cross-subject and cross-persona token distributions, revealing distinct lexical preferences and expression patterns under different Big Five settings. These differences

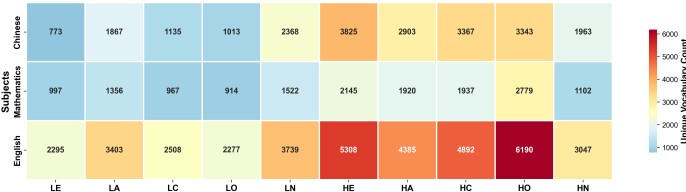

Figure 4: **Vocabulary richness across subjects and personas.** Virtual students show distinct vocabulary coverage across subjects and persona types, indicating that both dimensions substantially influence lexical diversity.

highlight the interpretability of persona-conditioned outputs and provide observable linguistic cues. Moreover, Fig. 4 presents a heatmap of vocabulary richness, showing that both subject domain and persona traits significantly affect lexical coverage. For example, English classes exhibit broader lexical diversity overall, while high extraversion and high openness personas consistently yield richer vocabulary across all subjects. These results demonstrate that the dataset not only achieves large scale and high consistency but also encodes quantifiable cross-subject and cross-persona linguistic variation, offering critical support for evaluation.

The final dataset thus achieves balanced coverage across subjects, languages, persona types, and behavior–expression dimensions. Its large scale and strong consistency make it well-suited for systematic evaluation of virtual student agents. Full distributions and examples are provided in the appendix for reproducibility and further extension.

## 4 EVALUATION FRAMEWORK

Evaluating virtual student agents poses the fundamental challenge of transforming inherently subjective abilities into measurable and reproducible statistics. To address this, we formalize the agent as a conditional generative model: $M : (x_t, C, p, s) \mapsto Y_t$, where $x_t$ denotes the teacher's input at turn $t$, $C$ the dialogue context (including prior turns), $p \in P$ the persona configuration of the student agent, $s \in S$ the subject domain, and $Y_t$ the generated response with associated labels. Equivalently, the model induces a conditional distribution $\mathbb{P}_M(Y_t \mid x_t, C, p, s)$, which is compared against a reference distribution $\mathbb{P}^\star(Y_t \mid x_t, C, p, s)$ derived from the dataset $\mathcal{D}$. The evaluation problem thus reduces to measuring the statistical divergence between $\mathbb{P}_M$ and $\mathbb{P}^\star$ under task-specific criteria. For experimental consistency, we partition the dataset $\mathcal{D}$ into a fine-tuning set $\mathcal{D}_{ft}$ and a held-out test set $\mathcal{D}_{test}$, following a fixed 6:4 split. Fine-tuning is performed exclusively on $\mathcal{D}_{ft}$, while all evaluations are conducted on $\mathcal{D}_{test}$. It ensures that reported results reflect genuine generalization performance rather than memorization of training data.

To make subjective ability assessment operational, we decompose the problem into three progressive tasks: (Task 1) **Basic coherence**: evaluating whether multimodal behavior–expression labels align with contextual semantics; (Task 2) **Student realism**: assessing whether responses follow authentic student traits and classroom norms (e.g., admitting ignorance, requesting hints, self-correcting); (Task 3) **Persona consistency**: examining whether persona traits and stylistic patterns remain stable across extended dialogues. These tasks form a progression from observable behaviors to perceived authenticity and long-term stability. All evaluations are stratified by subject $s$ and persona $p$, with confidence intervals and statistical tests ensuring robustness. In doing so, EduPersona transforms vague notions such as "realism" and "consistency" into quantifiable distributional properties, enabling systematic and reproducible benchmarking of virtual student agents. See App. C for details.

### 4.1 TASK 1: BASIC COHERENCE

At the most observable level, a virtual student agent should not only generate textual content but also produce behavior–expression patterns that are consistent with contextual semantics and the persona configuration. Given $(x_t, C, p, s)$, the model outputs a response $\hat{y}_t$ and a predicted behavior vector:

$\hat{b}_t = (\hat{beh}_t, \hat{emo}_t, \hat{exp}_t, \hat{voi}_t) \in B = B_{beh} \times B_{emo} \times B_{exp} \times B_{voi}$, where $B_{beh}$ (ICAP-based classroom behaviors), $B_{emo}$ (emotions), $B_{exp}$ (facial expressions), and $B_{voi}$ (vocal styles) form a *closed vocabulary*. Reference labels $b_t = (beh_t, emo_t, exp_t, voi_t)$ are generated by GPT-4 under persona and context constraints and then human-audited (see Sec. 3.2). We macro-average all metrics over the four dimensions, using the index set $\mathcal{I} = \{beh, emo, exp, voi\}$ and $T$ evaluation instances; $\emptyset$ denotes no output, and $B_i$ the codebook of dimension $i$.

**Step 1. Response rate.** We first check whether any output is produced on each dimension:

$$\text{RespRate} = \frac{1}{|\mathcal{I}|} \sum_{i \in \mathcal{I}} \frac{1}{T} \sum_{t=1}^{T} \mathbb{1}[\hat{b}_{t,i} \neq \emptyset]. \tag{1}$$

**Step 2. Validity rate.** Conditional on producing outputs, we verify whether they fall within the predefined vocabulary:

$$\text{ValidRate} = \frac{1}{|\mathcal{I}|} \sum_{i \in \mathcal{I}} \frac{\sum_{t=1}^{T} \mathbb{1}[\hat{b}_{t,i} \in B_i]}{\max\left(1, \sum_{t=1}^{T} \mathbb{1}[\hat{b}_{t,i} \neq \emptyset]\right)}. \tag{2}$$

**Step 3. Label prediction quality.** Restricting to dimensions with outputs, we evaluate agreement with the reference annotations. To disentangle different error sources, we define three complementary accuracies (same numerator—the number of correct labels—but different denominators):

$$\text{RawAcc} = \frac{1}{|\mathcal{I}|} \sum_{i \in \mathcal{I}} \frac{\sum_{t=1}^{T} \mathbb{1}[\hat{b}_{t,i} = b_{t,i}]}{\max\left(1, \sum_{t=1}^{T} \mathbb{1}[\hat{b}_{t,i} \neq \emptyset]\right)}, \tag{3}$$

$$\text{ValAcc} = \frac{1}{|\mathcal{I}|} \sum_{i \in \mathcal{I}} \frac{\sum_{t=1}^{T} \mathbb{1}[\hat{b}_{t,i} = b_{t,i}]}{\max\left(1, \sum_{t=1}^{T} \mathbb{1}[\hat{b}_{t,i} \in B_i]\right)}, \tag{4}$$

$$\text{OverallAcc} = \frac{1}{|\mathcal{I}|} \sum_{i \in \mathcal{I}} \frac{1}{T} \sum_{t=1}^{T} \mathbb{1}[\hat{b}_{t,i} = b_{t,i}]. \tag{5}$$

These three metrics satisfy the inequality $\text{OverallAcc} \leq \text{RawAcc} \leq \text{ValAcc}$, reflecting a progressive tightening from *availability* $\rightarrow$ *validity* $\rightarrow$ *end-to-end correctness*. Task 1 thus provides the most objective baseline for behavioral alignment and supplies interpretable low-level signals upon which higher-level evaluations of student realism (Task 2) and persona consistency (Task 3) can be built.

### 4.2 TASK 2: STUDENT REALISM

Student realism evaluates whether a model's response looks like a real student. This concept goes beyond linguistic fluency, requiring identity credibility (e.g., admitting ignorance, requesting hints, self-correction) and adherence to classroom interaction norms. To ground the evaluation, we consulted ten experts from education and AI, who reviewed a sampled subset of responses and distilled a set of core dimensions $\mathcal{R} = \{r_1, \ldots, r_m\}$, including linguistic naturalness, identity credibility, strategy appropriateness, and coordination with teacher feedback. These dimensions form the foundation for subsequent large-scale evaluation. We then encode $\mathcal{R}$ into prompts to construct an evaluation function: $G_{\mathcal{R}} : (x_t, C, p, s, \hat{y}_t) \longmapsto \{\hat{h}, \hat{\mathbf{z}}\}$, where $\hat{h} \in \{0, 1\}$ denotes the overall student-likeness decision and $\hat{\mathbf{z}} \in \{0, 1\}^m$ the dimension-wise outcomes. In this setup, GPT does *not* serve as an independent judge but as a scalable extension of the expert-derived criteria, ensuring interpretability and reproducibility. Results are aggregated across subjects $s$ and personas $p$, with macro-averaged scores reported at both overall and dimension levels, yielding a systematic, interpretable, and scalable evaluation of student realism.

### 4.3 TASK 3: PERSONA CONSISTENCY

Persona consistency requires virtual student agents to remain aligned with the target persona in both single-turn responses and extended dialogues. We define a standardized confidence function with range $[0, 1]$: $J(\hat{y}_t, p) \in [0, 1]$, where $J(\hat{y}_t, p) = 0$ indicates complete mismatch, $J(\hat{y}_t, p) = 1$ indicates perfect alignment, and intermediate values reflect partial consistency. Evaluation is conducted at two scales. For **short-term consistency**, each generated response $\hat{y}_t$ on the held-out

test set $\mathcal{D}_{test}$ receives a persona score, and the average across samples is reported. For **long-term consistency**, models engage in fixed 10-turn classroom-style interactions driven by a *Teacher-Policy* $\pi_T$ induced from the full dataset $\mathcal{D}$, which captures authentic instructional patterns such as IRF structures, scaffolding, and progressive difficulty. Scores are aggregated over the session to assess stability under sustained interaction. A unified metric is applied to both settings: $\text{Cons} = \frac{1}{|\mathcal{T}|} \sum_{t \in \mathcal{T}} J(\hat{y}_t, p)$, where $\mathcal{T} = \{1, \dots, N\}$ for short-term consistency (with $N$ test samples) and $\mathcal{T} = \{1, \dots, 10\}$ for long-term interactive sessions (fixed at 10 turns). This formulation allows us to examine both immediate persona alignment and its persistence throughout extended interactions.

## 5 EXPERIMENTAL DESIGN AND ANALYSIS

Building on EduPersona, we delineate the performance boundaries of virtual student agents across *basic coherence*, *student realism*, and *persona consistency*. We study two complementary settings in a unified protocol: baseline evaluation, where three representative foundation LLMs $\mathcal{M}_{\text{base}}^{(i)}$ are directly assessed without additional adaptation, and persona-conditioned evaluation, where each base model $\mathcal{M}^{(i)}$ is fine-tuned over the Big Five–based persona set $P = \{p_1, \dots, p_{10}\}$ to obtain $\mathcal{M}_{\text{ft}}^{(i)}(p)$, yielding $3 \times 10 = 30$ EduPersona-trained variants. All evaluations are run on the held-out test set $\mathcal{D}_{test}$, while fine-tuning uses only $\mathcal{D}_{ft}$; the complete dataset $\mathcal{D}$ is split 6:4 (train:test) to emphasize generalization. Within this setup, Sec. 5.1 outlines the model lineup and fine-tuning configuration; Secs. 5.2–5.4 report task-wise results. Further implementation details are provided in App. D.

### 5.1 EXPERIMENTAL ARCHITECTURE AND MODEL SELECTION

We employ three open-source foundation LLMs as bases: Qwen3-8B ($\mathcal{M}^Q$), noted for strong Chinese–English instruction following (Yang et al., 2025); InternLM3-8B-Instruct ($\mathcal{M}^I$), robust in Chinese educational scenarios with broad multilingual coverage (Cai et al., 2024); and DeepSeek-R1-Distill-Qwen-14B ($\mathcal{M}^D$), distilled for enhanced mathematical and logical reasoning (Guo et al., 2025). For each $\mathcal{M}^{(j)}$ and persona $p \in P = \{p_1, \dots, p_{10}\}$, we obtain a persona-specific variant $\mathcal{M}_{\text{ft}}^{(j)}(p)$, while the unadapted counterparts $\mathcal{M}_{\text{base}}^{(j)}$ serve as references. Fine-tuning follows a consistent LoRA configuration (rank $r = 16$, scaling $\alpha = 32$) with AdamW, learning rate $\eta = 3 \times 10^{-4}$, per-device batch size 8 and gradient accumulation 4, for up to 5 epochs. Each turn is encoded as a unified input $u_t = (x_t, C, p, s, b_t)$ to couple generative dialogue with closed-vocabulary behavior–expression codes. Inference settings and random seeds are aligned across models.

### 5.2 BASIC COHERENCE: CAN VIRTUAL STUDENTS GENERATE MULTIMODAL BEHAVIORS ALIGNED WITH CONTEXT?

Basic coherence requires virtual students not only to produce text but also to align behaviors, emotions, expressions, and vocal styles with the classroom context. We evaluate three model families using the five metrics defined in Sec. 4.1—response rate, validity rate, RawAcc, ValAcc, and OverallAcc—macro-averaged across four dimensions (behavior, emotion, expression, voice).

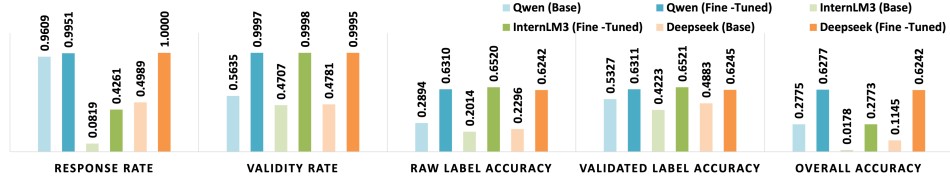

Figure 5: **Effect of persona fine-tuning on basic coherence.** Fine-tuning consistently enhances all five coherence metrics across models. Qwen and DeepSeek exceed 0.62 OverallAcc with strong label alignment, while InternLM3 also improves but remains limited by its lower response rate.

Fig. 5 shows that **persona fine-tuning substantially enhances multimodal alignment**. Qwen (0.9951/0.9997) and DeepSeek (1.0000/0.9995) achieve near-perfect response and validity rates, while InternLM3 retains high validity (0.9998) but suffers from low response coverage (0.4261). For label alignment, fine-tuned RawAcc and ValAcc converge in the 0.624–0.653 range, a large improvement over base versions (e.g., Qwen: 0.2894 → 0.6310). With validity nearly saturated,

Raw and Val scores converge, whereas base models show wider Raw–Val gaps due to frequent OOV outputs (e.g., DeepSeek: $0.2296 \rightarrow 0.4883$). For end-to-end correctness, Qwen and DeepSeek reach OverallAcc of 0.6277 and 0.6242, far above their baselines (0.2775/0.1145), while InternLM3 remains at 0.2773, constrained by limited response coverage.

In summary, Task 1 demonstrates that persona fine-tuning substantially improves basic coherence: Qwen and DeepSeek approach practical levels of multimodal alignment, while InternLM3 remains constrained by response generation. Beyond the overall gains, the fine-grained results in App. D.1 reveal two key insights. First, a consistent difficulty hierarchy emerges across dimensions (Emotion being the easiest and Behavior the hardest) indicating that residual errors arise from discourse- and intent-level challenges rather than random noise. Second, Qwen and DeepSeek converge to a similar end-to-end accuracy after fine-tuning (OverallAcc $\approx 0.62$), implying that task structure and dataset design, rather than model scale alone, largely determine the performance ceiling. With validity already near saturation, future improvements are likely to depend on expanding response coverage and strengthening fine-grained behavioral guidance.

## 5.3 STUDENT REALISM: CAN VIRTUAL STUDENTS BE LIKE REAL STUDENTS?

Student realism is a key criterion for evaluating whether virtual students resemble real learners. Using expert-derived dimensions operationalized through an automatic evaluator, we analyze results from both persona and model perspectives (Fig. 6).

**Persona-wise analysis (Fig. 6a).** Fine-tuned models exhibit consistently stronger realism across all personas, yet heterogeneity persists. High Conscientiousness (HC) and High Openness (HO) remain lower both before and after fine-tuning, with modest gains, whereas High Neuroticism (HN), Low Conscientiousness (LC), and Low Openness (LO) achieve relatively high realism. This reflects the interaction between persona traits and model defaults: HC/HO emphasize objective, comprehensive, and teacher-aligned behaviors, overlapping with LLMs' inherent answer-first tendency, thus appearing more machine-like. By contrast, HN/LC/LO manifest hesitation, partial responses, or self-corrections, which enhance perceived authenticity and yield higher realism.

**Model-wise analysis (Fig. 6b).** At the baseline level, the three families differ substantially (Qwen 0.7019, InternLM3 0.6478, DeepSeek 0.556). After fine-tuning, however, all models converge to a narrow band around 0.82 (Qwen 0.8221, InternLM3 0.8265, DeepSeek 0.8176). This demonstrates that persona conditioning both elevates student realism and reduces inter-model disparities.

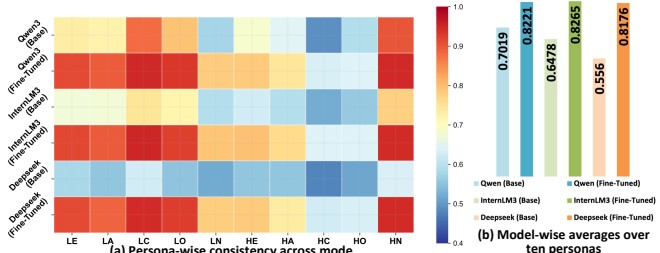

Figure 6: **Persona fine-tuning markedly improves student realism.** EduPersona brings all three model families to a converged level around 0.82 while revealing stable persona-specific differences (HC/HO harder, HN/LC/LO easier).

In summary, Task 2 demonstrates that persona fine-tuning markedly improves student realism and brings different model families to a converged performance level, although persona-specific bottlenecks persist. As detailed in App. D.2, gains are uneven across models—with DeepSeek benefiting the most—and cross-task regularities emerge: the Emotion dimension (easiest in Task 1) aligns with high-realism personas (HN/LN/LO), while the Behavior dimension (hardest in Task 1) aligns with low-realism personas (HC/HO). These patterns affirm EduPersona's ability to enhance higher-level perceptual realism while revealing trait-specific challenges, forming a strong foundation for evaluating long-horizon persona consistency.

## 5.4 PERSONA CONSISTENCY: CAN VIRTUAL STUDENTS MAINTAIN STABLE PERSONAS DURING INTERACTIONS?

Task 3 evaluates whether virtual students can maintain stable persona traits throughout interactions. Results show that fine-tuned models achieve substantially higher and more stable consistency at both persona and model levels, confirming the effectiveness of persona conditioning.

**Persona-wise analysis (Fig. 7a).** While fine-tuning consistently improves all ten personas, heterogeneity persists. High Conscientiousness (HC, 0.731) and High Openness (HO, 0.779) remain the most difficult to sustain, even after adaptation, whereas High Neuroticism (HN, 0.901), Low Conscientiousness (LC, 0.887), and Low Openness (LO, 0.873) achieve the highest stability. This echoes Task 2's findings on realism, suggesting that structured, "idealized" personas are both less authentic and less consistent, while personas reflecting hesitation or partial responses are easier to maintain. Gains also vary across personas: Low Extraversion (+0.146), Low Openness (+0.133), and Low Agreeableness (+0.130) benefit most, whereas High Openness (+0.062) and High Conscientiousness (+0.073) improve the least.

**Model-wise analysis (Fig. 7b).** At the baseline level, Qwen, InternLM3, and DeepSeek differ significantly (0.795, 0.723, and 0.677). After fine-tuning, however, all three converge to a narrow range of 0.833–0.841, showing that persona conditioning boosts consistency while reducing cross-family disparities.

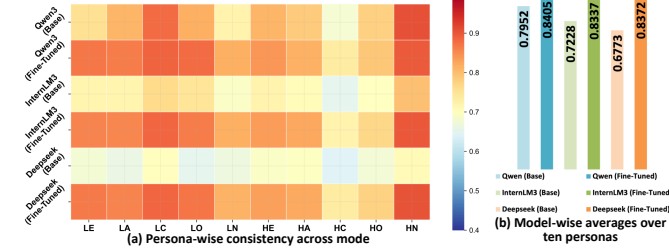

**Multi-turn consistency.** To compare the persona retention ability of fine-tuned versus closed-source models, we conducted a 10-turn English classroom experiment, involving three LoRA-fine-tuned models (Qwen3, InternLM3, DeepSeek) and GPT-4o. LoRA-fine-tuned models retained persona traits even after explicit prompts disappeared, with Qwen3-LoRA achieving the highest overall score ($0.920 \pm 0.042$). All three models maintained $\geq 0.8$ in later turns (6–10), whereas GPT-4o quickly drifted once persona instructions vanished, averaging only $0.480 \pm 0.262$ and dropping below 0.50 in later turns. This highlights the importance of fine-tuning for long-horizon persona stability.

Figure 7: **Persona fine-tuning enhances persona stability.** Fine-tuned models converge around 0.84 while showing consistent persona-specific differences (HC/HO harder, HN/LC/LO easier).

In summary, Task 3 shows that persona consistency is more demanding than student realism, yet persona fine-tuning substantially improves long-horizon stability and reduces both model-level and trait-level disparities. Persistent bottlenecks remain for HC and HO personas, whereas HN, LC, and LO are consistently easier to maintain. Full score distributions and supplementary analyses are provided in App. D.3.

## 5.5 RELIABILITY VALIDATION OF AUTOMATED METRICS

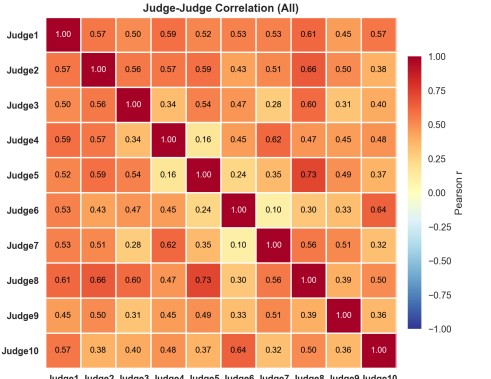

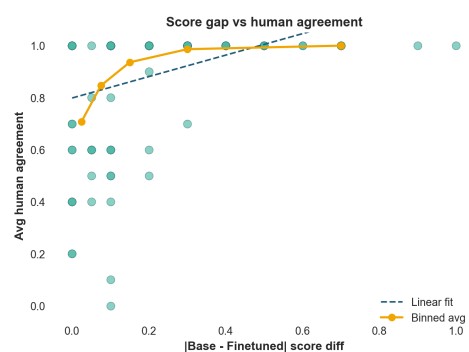

(a) Judge–judge correlation matrix

(b) Score gap vs. human agreement

Figure 8: **Reliability of subjective evaluation.** (a) Inter-annotator correlations reveal a stable and structured agreement pattern across 10 experts, rather than random variability. (b) Human–AI agreement increases monotonically with model score gaps, reaching full consensus at large performance differences, demonstrating the evaluator's sensitivity and alignment with human judgment.

To validate the reliability of the subjective metrics in Task 2 and Task 3, we constructed 120 evaluation instances spanning three model families and ten persona types, and collected 1,200 blind judgments from 10 experts.

**Inter-judge consistency**. As shown in Fig. 8a, pairwise Pearson correlations predominantly fall within 0.40–0.60, with several pairs reaching 0.60–0.73. The absence of low or negative correlations indicates a stable and structured agreement pattern rather than random variability, providing a reliable human reference for evaluating automated metrics.

**Human–AI alignment**. Building on this stable human baseline, the automated evaluator achieves strong alignment with expert consensus: 95.0% majority agreement on Task 2, 90.0% on Task 3, and 92.5% overall. Given the inherently subjective nature of the tasks, such alignment indicates that the automated metrics capture the core decision logic used by experts.

**Sensitivity to performance differences**. As shown in Fig. 8b, human agreement increases monotonically with the score gap between base and fine-tuned models. Agreement is around 0.7 when the gap is small ($\leq 0.05$), rises sharply to 93–100% for moderate gaps (0.10–0.40), and reaches 100% for large gaps (>0.40). This demonstrates that the automated evaluator not only produces stable judgments but also faithfully reflects human sensitivity to model performance differences.

Overall, the experts exhibit a clear and stable consistency structure, and the automated evaluator aligns closely with both their overall decisions and their sensitivity patterns, validating GPT-4o as a reliable, sensitive, and interpretable tool for subjective ability assessment. For a comprehensive breakdown of the experimental setup, sensitivity analysis, and fine-grained reliability across models, please refer to App. E.

## 5.6 SUMMARY

The three experiments outline the capability landscape of virtual student agents. In **Task 1 (basic coherence)**, Qwen and DeepSeek converge around 0.62 after fine-tuning (0.6277/0.6242), while InternLM3 remains substantially lower (0.2773) due to limited response coverage (RespRate 0.4261; App. D.1). For **Task 2 (student realism)**, initial gaps (Qwen 0.7019, InternLM3 0.6478, DeepSeek 0.5560) largely disappear after fine-tuning, with all models converging near 0.82 (App. D.2). In **Task 3 (persona consistency)**, scores rise from 0.795/0.723/0.677 to ∼0.84 (0.841/0.834/0.837), again showing strong gains and cross-family convergence (App. D.3). Together, these results show that EduPersona improves absolute performance while aligning outcomes across model families. The tasks form a clear progression: Task 1 evaluates structural compliance, Task 2 human-perceived realism, and Task 3 long-horizon trait stability. Persona-level patterns are consistent—**HC/HO persist as bottlenecks**, whereas **HN/LC/LO are easier to emulate**, mirroring Task 1's "Emotion easier, Behavior harder" trend. A 10-turn analysis further shows that LoRA-fine-tuned models maintain persona traits even when prompts vanish (e.g., Qwen3-LoRA 0.920±0.042), whereas GPT-4o exhibits drift (0.480±0.262), indicating that **long-range stability depends on fine-tuning rather than model scale**. Overall, performance ceilings are shaped more by persona type and interaction length than by parameter count, motivating future work on improving coverage, refining behavior supervision, and incorporating hesitation–repair–collaboration signals into idealized personas.

## 6 CONCLUSION

This work presents **EduPersona**, a systematic framework for evaluating three core subjective abilities of virtual student agents—basic coherence, student realism, and persona consistency. Experimental results show that persona fine-tuning substantially improves performance, with models converging to scores of 0.62, 0.82, and 0.84 across the three tasks. Stable persona-specific gaps remain: traits such as high conscientiousness and openness are harder to emulate, whereas high neuroticism, low conscientiousness, and low openness are easier to sustain. These patterns reveal structural links between personality traits and model behaviors. We further conduct a supplementary human–AI alignment study demonstrating that GPT-4o's judgments closely match expert consensus, providing strong empirical support for the reliability of the automated evaluation metrics. Overall, EduPersona outlines a progression from structural compliance to perceived realism and long-horizon trait stability, highlighting the central role of behavioral dimensions and sustained dialogue. While fine-tuning narrows performance disparities, challenges persist in response coverage, idealized personas, and long-term stability. Future work should strengthen fine-grained behavioral supervision, promote more naturalistic interaction, and extend the framework to broader interdisciplinary settings, positioning EduPersona as a pathway toward trustworthy and human-aligned educational agents.

## ETHICS STATEMENT

This work complies with ethical standards and data usage regulations. The EduPersona dataset is constructed from publicly available classroom resources, with all data anonymized to remove IDs and other identifiers. Persona configurations are derived from the well-established Big Five personality theory and serve only as abstract stylistic constructs for modeling language and behavior; they do not encode demographic or sensitive attributes. Thus, this study introduces no new ethical risks and instead provides a safe, controlled, and reproducible framework for evaluating virtual students in educational contexts.

## REPRODUCIBILITY STATEMENT

To ensure reproducibility, we will open-source the complete codebase, annotation guidelines, preprocessing pipeline, and trained models. The released EduPersona resources will include the full data schema (prompts, vocabularies, labels, and evaluation metrics). Third-party raw media will not be redistributed due to licensing restrictions; instead, we provide scripts to re-derive processed text from public resources. Appendix B details data collection and preprocessing, Appendix B.4.1 and B.4.2 describe persona configurations and stylization prompts, and Appendix C provides the evaluation prompts. Additional fine-grained analyses are reported in Appendix D.1–D.3 to further substantiate our conclusions. All code, data, and models will be released upon acceptance to foster long-term reproducibility and extensibility in educational agent research.

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

APPENDIX

This appendix provides implementation details of the EduPersona corpus and the processing steps that support the evaluation framework in the main paper, along with additional fine-grained experimental results that further substantiate our conclusions.

## A    THE USE OF LARGE LANGUAGE MODELS (LLMs)

During manuscript preparation, Large Language Models (LLMs) were employed solely for language refinement and stylistic polishing.

## B    DATASET DETAILS

### B.1    DETAILED DATA COLLECTION AND PREPROCESSING

The EduPersona corpus is constructed from publicly accessible classroom resources licensed for research, covering three subjects (Chinese, Mathematics, English) across two languages (Chinese/English). Sources include: (i) curated Chinese open-class videos and transcripts from a national education platform, (ii) international mathematics discourse corpora, and (iii) English classroom transcripts. Only segments with clear instructional intent are retained; all personally identifiable information is masked, and collection complies with the original platforms' terms of use. No redistribution of third-party raw media is performed.

**Preprocessing and structuring.** A unified pipeline ensures cross-subject comparability. Videos and subtitles (or ASR transcripts where subtitles are unavailable) are aligned at the utterance level with normalized punctuation and casing. Teacher–student roles and IRF structures are reconstructed, narration and meta-comments are removed, and turn boundaries are adjusted to avoid truncation. Identifiers such as IDs are masked at source to guarantee non-attributability. A human-in-the-loop process (automatic tagging, sampled review, manual correction) yields verified role labels.

**Persona expansion and behavior labeling.** To enable persona-conditioned dialogue modeling, each student response is expanded into ten variants using the Big Five dimensions $\mathcal{F} =$ Extraversion, Agreeableness, Conscientiousness, Neuroticism, Openness with high/low polarity, giving $P = \mathcal{F} \times H, L$. Each original $(x, y)$ is rewritten into $y^{(p)}$ that preserves semantics while adapting to persona $p$. In addition, each $(x, y^{(p)})$ is labeled with a four-dimensional vector $b = (beh, emo, exp, voi)$ covering *Behavior*, *Emotion*, *Expression*, and *Voice*, constrained to a fixed vocabulary (App. B.4.3). Low-confidence or contradictory cases are resolved through human auditing.

**Split, statistics, and quality control.** The full dataset $\mathcal{D}$ is stratified by subject and persona into fine-tuning ($\mathcal{D}ft$) and test sets ($\mathcal{D}test$) with a 6:4 ratio. All evaluation tasks (Tasks 1–3) are performed exclusively on $\mathcal{D}test$. Before persona expansion, the base corpus $\mathcal{D}base$ contains **1,308** dialogue rounds with **12,814** teacher–student Q&A turns. By subject: Chinese contributes 401 rounds from 32 lessons (1,531 Q&A turns; avg. 3.82/round), Mathematics 395 rounds (3,420 Q&A turns; avg. 8.66/round), and English 512 rounds (7,863 Q&A turns; avg. 15.36/round). After stylization, each turn yields ten persona-conditioned variants, expanding the dataset tenfold. Quality control is embedded at every stage: role integrity verified against source materials, persona fidelity checked by independent judges, label validity enforced with 100% in-vocabulary coverage, and near-duplicates removed via semantic similarity. These measures ensure that EduPersona is reliable, diverse, and ethically compliant, providing a solid foundation for the evaluation framework and subsequent experiments.

### B.2    DATA QUALITY VERIFICATION: LABEL VALIDITY

To rigorously assess the validity of the four multimodal label dimensions—behavior, emotion, expression, and voice—we conducted a post-hoc human audit using a strict blind-review protocol. We randomly sampled 20 dialogue instances covering all 10 Big Five persona types and invited 10 domain experts (with backgrounds in education or behavioral analysis) to independently judge the reasonableness of the labels, yielding 200 binary (reasonable / unreasonable) evaluations. This design directly examines the authenticity of the annotations and their cross-observer consistency.

**High Reliability and Consensus.** As shown in Table A1, the labels exhibit strong overall reliability: the average human acceptance rate is **90.5%**, with individual judges ranging from **85% to 95%**. Majority approval reaches **90%**, and **85%** of the samples receive unanimous agreement (10/10 votes)—a remarkably high rate for fine-grained subjective judgments. Further analysis reveals that **8 of the 10 persona types achieve 100% acceptance**, and **17 of the 20 samples** receive unanimous approval, indicating that neither persona variation nor sample content meaningfully affects label quality.

Table A1: Human Verification of Dataset Labels

| Metric | Value |
|---|---|
| Average Acceptance Rate | 90.5% |
| Majority Agreement Rate ($> 50\%$) | 90.0% |
| Full Agreement Rate (100%) | 85.0% |

**Inter-Annotator Consistency.** Fig. A1(Left) shows that all judges maintain acceptance rates above 85%, with no outliers. The pairwise correlation matrix in Fig. A1(Right) further indicates strong alignment across evaluators, with an average Pearson correlation of **0.74**, well above typical values for subjective annotation tasks. This confirms that expert judgments follow a stable, shared standard rather than random variation.

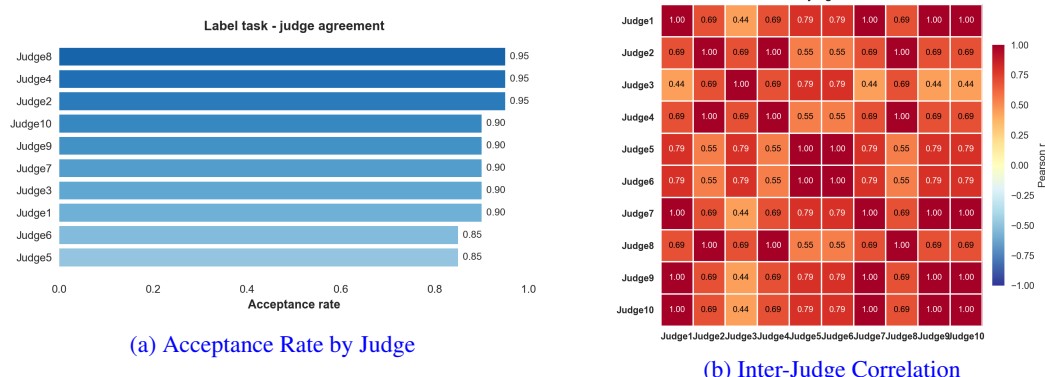

(a) Acceptance Rate by Judge

(b) Inter-Judge Correlation

Figure A1: **Reliability of Label Verification.** (a) Individual acceptance rates for the label audit task, all exceeding 85%. (b) Pairwise correlation matrix for label judgments, showing high consistency across experts.

Importantly, the evaluation criteria were intentionally strict: a sample was marked as "not fully accepted" if *any* of its four dimensions appeared questionable. Achieving 90.5% acceptance and 85% unanimity under such stringent conditions demonstrates the robustness of our annotation pipeline. The absence of perfect consensus reflects the inherent complexity of multimodal subjective judgment, rather than any deficiency in the labeling process.

### B.3  DATA ACCESS AND ETHICS

We will release full preprocessing and labeling code, along with a legally redistributable subset that mirrors the complete schema (including prompts, vocabularies, and metric definitions), thereby supporting reproducibility of all reported experiments. Third-party raw media are not redistributed; instead, scripts are provided to re-derive processed text from publicly available resources where permitted.

All dialogues are fully de-identified, with names, IDs, and any sensitive information removed. Persona variants are abstract stylistic constructs derived from the Big Five framework and do not encode demographic attributes. We explicitly caution against any attempt at re-identification or demographic inference. The EduPersona corpus is designed under principles of compliance, transparency, and responsible use, ensuring safe deployment in educational and AI research contexts.

## B.4 MORE INFORMATION ABOUT ANNOTATION

### B.4.1 TEN PERSONA CONFIGURATIONS BASED ON BIG FIVE THEORY

Our persona framework is grounded in the Big Five personality theory (BFAS scale), extending it into ten standardized configurations that represent authentic student behavioral patterns in classroom settings. Each persona is defined through multiple dimensions to ensure comprehensive and scientifically-based characterization.

Table A2: Ten persona configurations based on Big Five personality theory.

| Persona Type | Detailed Characteristics |
|---|---|
| **High Extraversion** | **Core traits**: Take charge, have a strong personality, warm up quickly to others
**Behavioral traits**: Active participation, strong social skills, comfortable expression, high exhibition, obvious dominance
**Language style**: Fluent and confident, detailed elaboration, positive attitude, clear structure, likes to lead conversations
**Classroom behavior**: Strong desire to participate, actively answers questions, proactively shows themselves, frequent interaction with teachers
**Response patterns**: Detailed answers with additional explanations, fluent language but may include repetitions and filler words |
| **Low Extraversion** | **Core traits**: Do not have an assertive personality, hard to get to know, keep others at a distance
**Behavioral traits**: Introverted and cautious, doesn't like to show off, socially conservative, low participation
**Language style**: Concise and conservative, cautious responses, avoids excessive expression, passive responses
**Classroom behavior**: Less proactive participation, speaks softly, doesn't want to be center of attention
**Response patterns**: Brief answers, sometimes needs teacher encouragement to speak, lighter tone |
| **High Agreeableness** | **Core traits**: Sympathize with others' feelings, take an interest in other people's lives, feel others' emotions
**Behavioral traits**: Cooperative and caring, considerate and patient, positive feedback, understanding, high empathy
**Language style**: Gentle and friendly, detailed elaboration, accurate expression, caring for others
**Classroom behavior**: Happy to help classmates, actively participates in discussions, pays attention to others' feelings
**Response patterns**: Gentle language, accurate expression, rarely makes mistakes, considers others' feelings |
| **Low Agreeableness** | **Core traits**: Can't be bothered with others' needs, take advantage of others, not interested in other people's problems
**Behavioral traits**: Competitive, direct and frank, self-focused, less compromising, lacks empathy
**Language style**: Direct and clear, sometimes slightly harsh, focuses on personal views
**Classroom behavior**: More focused on own performance, may lack patience with others' opinions
**Response patterns**: Direct answers, sometimes slightly harsh, more focused on expressing own views |
| **High Conscientiousness** | **Core traits**: Keep things tidy, like order, carry out plans
**Behavioral traits**: Serious and responsible, organized, clear goals, persistent, highly self-disciplined |

| | Table A2 continued from previous page |
| --- | --- |
| **Persona Type** | **Detailed Characteristics** |
| | **Language style**: Clear and logical, accurate and error-free, strong logic, complete expression
**Classroom behavior**: Listens carefully, well-prepared when answering questions, solid grasp of content
**Response patterns**: Accurate and complete answers, clear logic, rarely makes mistakes |
| **Low Conscientiousness** | **Core traits**: Waste time, find it difficult to get down to work, leave belongings around
**Behavioral traits**: Careless, poor organization, easily distracted, lacks persistence
**Language style**: Simple and direct, occasional errors, incomplete expression, sometimes inconsistent
**Classroom behavior**: Easily distracted, unprepared when answering questions
**Response patterns**: Unstable answers, sometimes right sometimes wrong, incomplete expression |
| **High Neuroticism** | **Core traits**: Get upset easily, get angry easily, get easily agitated
**Behavioral traits**: Nervous and anxious, emotional fluctuations, sensitive and irritable, emotionally unstable
**Language style**: Hesitant and indecisive, repetitive backtracking, incoherent expression, full of uncertainty
**Classroom behavior**: Easily nervous, sensitive to classroom environment, shows worry when answering
**Response patterns**: Full of 'um', 'uh' filler words, repeats and backtracks, incoherent expression |
| **Low Neuroticism** | **Core traits**: Rarely get irritated, not easily annoyed, feel comfortable with self
**Behavioral traits**: Emotionally stable, calm and composed, strong stress adaptability, high self-acceptance
**Language style**: Stable and natural, clear logic, calm expression, few emotional fluctuations
**Classroom behavior**: Remains calm when facing problems, peaceful attitude when answering
**Response patterns**: Stable and natural answers, clear logic, shows inner calm and confidence |
| **High Openness** | **Core traits**: Quick to understand things, believe in the importance of art, can handle a lot of information
**Behavioral traits**: Strong curiosity, imaginative, accepts new things, flexible thinking, values aesthetics
**Language style**: Creative and rich expression, good at association, broad thinking
**Classroom behavior**: Curious about new knowledge, good at asking questions, creative answers
**Response patterns**: Creative answers, makes associations and extensions, can handle complex information |
| **Low Openness** | **Core traits**: Have difficulty understanding abstract ideas, do not like poetry, seldom notice emotional aspects of art
**Behavioral traits**: Conservative and traditional, relies on experience, low acceptance, rigid thinking
**Language style**: Simple and direct, lacks extension, difficulty going deep, prefers concrete descriptions |
| | Continued on next page |

**Table A2 continued from previous page**

| Persona Type | Detailed Characteristics |
|---|---|
| | **Classroom behavior**: Tends to rely on existing knowledge, low acceptance of new content |
| | **Response patterns**: Simple and direct answers, lacks extension, prefers concrete answers over abstract analysis |

### B.4.2 PROMPT TEMPLATE FOR PERSONA STYLIZATION

To implement personality-driven dialogue generation, we design a structured prompt template that formalizes the subjective task of persona stylization into a systematic workflow. The system prompt defines the model as an "expert in Big Five personality theory" and establishes rules that govern three major components:

First, **student speech processing** is grounded in the five BFAS dimensions, with placeholders specifying core traits, behavioral tendencies, linguistic patterns, classroom manifestations, and response styles. This multi-dimensional mapping ensures that student responses preserve semantic meaning while being rewritten into styles consistent with the target persona.

Second, **teacher speech processing** follows a conservative strategy: original content is preserved whenever possible, with minimal modifications applied only in four predefined cases (e.g., coherence or disambiguation). This guarantees that instructional intent remains intact.

Finally, **quality and output requirements** enforce strict formatting rules and consistency checks, balancing stylistic fidelity with semantic preservation. By integrating these layers, EduPersona transforms persona-conditioned dialogue generation into a reproducible and scalable process, enabling consistent expansion and evaluation in large-scale educational applications.

---

**The Prompt for Persona Stylization.**

**System Prompt**
You are an expert deeply knowledgeable in student psychology and Big Five personality theory, particularly skilled at simulating different personality traits of students in classroom settings based on the Big Five personality framework.
Your task is to regenerate student speech that conforms to specific personality traits based on provided real teacher-student dialogues, while handling teacher speech according to strict rules.
**##INSTRUCTIONS:**
- Student speech processing principles, reflecting target student personality traits - [TARGET PERSONALITY]:

- Core traits: [CORE TRAITS FROM BFAS SCALE]

- Behavioral traits: [BEHAVIORAL CHARACTERISTICS]

- Language style: [LANGUAGE PATTERNS]

- Classroom behavior: [CLASSROOM MANIFESTATIONS]

- Response patterns: [TYPICAL RESPONSE STYLES]

- Audio clarity handling: When encountering unclear or inaudible portions, intelligently infer and complete missing content based on dialogue context, logical flow, and target personality traits

- Teacher speech processing principles:

1. Keep teacher's original words unchanged by default

2. Only make adjustments in the following situations:

- Teacher's referential content doesn't match student's new response
- Teacher's follow-up questions don't match student's new response structure
- Teacher's guidance obviously conflicts with student's new state
- Teacher's speech contains unclear or inaudible portions

---

> 3. When modifying, only adjust referential content and connection logic, maintain teacher's educational intent and professionalism
>
> 4. Audio clarity handling: When encountering unclear or inaudible portions in teacher's speech, intelligently infer and complete the missing content based on educational context, teaching objectives, and professional pedagogical patterns
>
> - Important requirements:
>
> 1. Strictly follow the scientific descriptions in the prompts to shape student personality traits, and student personality traits should remain consistent throughout the dialogue
>
> 2. Student responses should match appropriate student knowledge level and vocabulary
>
> 3. Maintain the educational significance and logical relationships of the dialogue
>
> 4. For unclear or inaudible portions in the original dialogue (both teacher and student speech), use contextual inference and appropriate behavioral patterns to complete missing content naturally
>
> - **DO NOT PROVIDE ANY OTHER OUTPUT TEXT OR EXPLANATION**. Output strictly in specified format without explanatory text.
>
> ---
>
> **User:**
> Original teacher-student dialogue: [ORIGINAL DIALOGUE TEXT]
> Please regenerate the dialogue based on the above information, where:
> **Student speech requirements:**
>
> - Completely regenerate according to [TARGET PERSONALITY] personality traits
> - Reflect typical performance and response patterns of this personality in classroom settings
> - Maintain appropriate student knowledge level and expression style
> - For any unclear portions, infer and complete based on context and personality traits
>
> **Teacher speech requirements:**
>
> - Prioritize keeping original words unchanged
> - Only make adjustments when references don't match, follow-ups don't align, emotions conflict, or speech is unclear
> - For unclear portions, infer and complete based on educational context and teaching objectives
> - Maintain teaching objectives and professional expression unchanged
>
> Please strictly follow the format below for output, maintain the same number of dialogue turns as the original dialogue, without any other explanatory text:
> Teacher: [Teacher's words]
> Student: [Student's words]
> Teacher: [Teacher's words]
> Student: [Student's words]
> ...

### B.4.3 BEHAVIOR–EXPRESSION LABEL SPACE

The behavior–expression annotation system employs a controlled vocabulary across four dimensions.

Table A3: Behavior–expression controlled vocabulary with operational definitions.

| Dimension | Label | Operational Definition |
|-----------|-------|------------------------|
| | Simple Response | Answers with "yes/no", "I don't know". |

**Behavior**

**Table A3 (continued)**

| Dimension | Label | Operational Definition |
|---|---|---|
| | Mechanical Repetition | Repeats the teacher's question or content verbatim. |
| | Standing Answer | Independently provides a complete answer to the teacher's question. |
| | Example Explanation | Actively uses examples to explain knowledge. |
| | Summary Generalization | Summarizes the learned content; expresses personal understanding. |
| | Active Questioning | Asks questions to express confusion or reflective thoughts. |
| | Supplementary Speech | Expands or supplements others' viewpoints. |
| | Opinion Expression | Refutes or negotiates with others' statements. |
| **Emotion** | Positive | Positive/engaged/confident state. |
| | Confused | Unclear thinking or partial understanding but still trying. |
| | Negative | Disappointed, frustrated, helpless emotions. |
| **Expression** | Smile | Pleasant, satisfied facial expression. |
| | Frown | Confused or dissatisfied facial expression. |
| | Nod | Agreeing/understanding/affirming action. |
| | Shake Head | Disagreeing or confused action. |
| | Silence | No obvious facial change; remains calm. |
| **Voice** | Hesitant | "um/uh/I think/maybe", disfluent. |
| | Confident | Clear, continuous, steady tone. |
| | Enthusiastic | Higher pitch, emotional emphasis. |
| | Low-tone | Low and weak voice. |
| | Neutral | Medium rate and tone; no strong affect. |

## C  EVALUATION PROMPT TEMPLATE FOR TASK 2 & 3

Our evaluation adopts a dual-task framework combining automatic language detection with adaptive prompting to systematically measure *dialogue realism* (Task 2) and *persona consistency* (Task 3). This design ensures comparability and reproducibility across multilingual datasets.

---

**Prompt Template for Dialogue Evaluation (GPT-4o)**

**System Prompt**
You are a professional dialogue evaluation expert, skilled in analyzing dialogue realism and persona consistency. Return the evaluation strictly in JSON format.

---

**User Prompt Structure**
**Dialogue Context:** [FULL DIALOGUE CONTENT]
**Student's Final Response to Evaluate:** "[STUDENT'S LAST RESPONSE]"
**Target Persona Description:** "[TARGET PERSONALITY TRAITS DESCRIPTION]"
**Evaluation Tasks**
**Task 2 – Realism Assessment** Evaluate how realistic and natural the student's final response is, in line with human conversational characteristics.
*Guidelines:*

- Focus exclusively on the final response.
- Consider fluency, logical coherence, natural emotional expression; penalize mechanical or overly structured replies.
- Strictly penalize textbook-like or formalized answers.
- Response length is irrelevant: short, casual replies can be more realistic.

*Scoring Scale:*

- 0.9–1.0: Perfectly natural, conversational, spontaneous
- 0.7–0.8: Mostly natural, minor formal elements
- 0.5–0.6: Partly mechanical or structured
- 0.3–0.4: Clearly artificial or textbook-like
- 0.0–0.2: Completely mechanical or unrealistic

**Task 3 – Persona Consistency Assessment** Evaluate how naturally and realistically the student's final response reflects the *target persona traits*, **relative to the provided persona description**.
*Guidelines:*

- Natural expression outweighs exaggerated or templated trait markers.
- Consistency must be judged with reference to the provided persona description.
- Consistency does not require repetition or extremity.
- Penalize repetitive, formulaic, or mechanized expressions.
- Allow minor deviations; focus on overall alignment.
- Length is not decisive; natural reflection of traits is key.

*Scoring Scale:*

- 0.9–1.0: Natural and realistic reflection of traits, no templated artifacts
- 0.7–0.8: Mostly consistent, slight templating
- 0.5–0.6: Partial consistency, mixed traits, less natural
- 0.3–0.4: Traits weak or inconsistent, but still natural
- 0.0–0.2: Contradicts traits and unnatural

**Important:** Evaluate only the final student response. Do not use previous dialogue turns as reference.
**Expected Output (strict JSON):**

---

```
{
    "task2_realism": {"score": 0.0-1.0},
    "task3_persona": {"score": 0.0-1.0}
}
```

This framework ensures robustness and reproducibility through three mechanisms. First, language is automatically detected via character-level statistics, enabling adaptive switching between Chinese and English prompts with corresponding persona descriptions. Second, a retry mechanism with exponential backoff and strict JSON parsing with fallback patterns safeguard evaluation reliability. Third, the dual-task design balances realism and persona consistency, scoring only the student's final response to maintain contextual grounding while ensuring comparability across scenarios.

Overall, this evaluation framework transforms subjective persona assessment into a structured, replicable process, providing the methodological foundation for large-scale experiments and statistical analysis.

# D  DETAILED EXPERIMENTAL RESULTS

## D.1  ADDITIONAL ANALYSIS FOR TASK 1

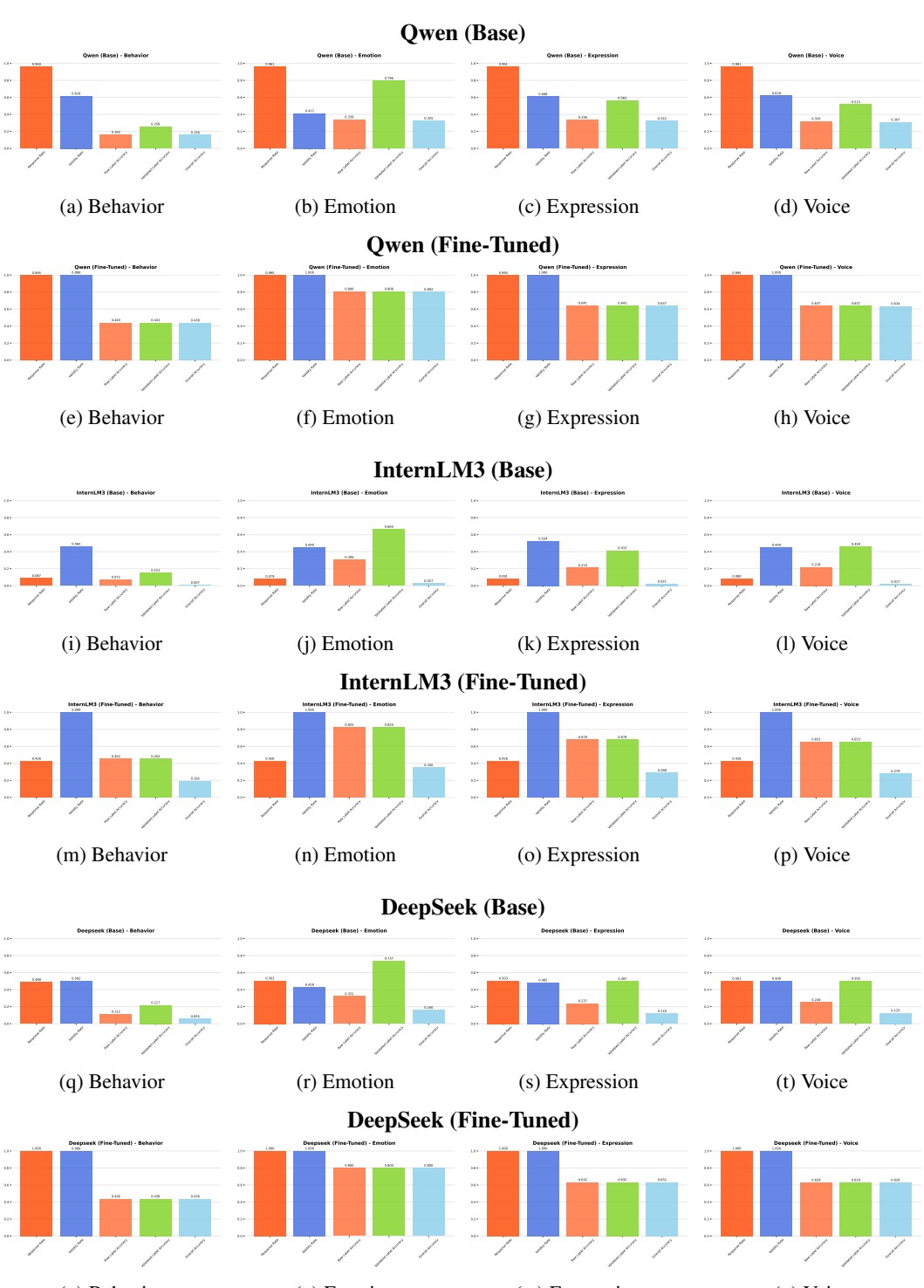

Figure A2: **Task 1 — All model families (24 charts).** Weighted averages over ten personas; five metrics per chart for each model family (Qwen, InternLM3, DeepSeek) showing both base and fine-tuned variants across four dimensions (Behavior, Emotion, Expression, Voice).

Fig. A2 presents a comprehensive view of Task 1 across three model families, covering both base and fine-tuned settings, four behavioral dimensions, and five evaluation metrics. This integrated view makes it possible to trace how fine-tuning improves performance at both the overall and dimension-specific levels.

The end-to-end gains can be understood as the combined effect of three factors: whether the model produces an output at all (response coverage), whether the output is structurally valid (vocabulary compliance), and whether the valid outputs align correctly with the reference labels (in-vocabulary discrimination). For Qwen and DeepSeek, structural validity is already saturated after fine-tuning, so their improvements mainly come from higher label alignment together with strong response coverage. By contrast, InternLM3 also benefits from improved label alignment, but its relatively low response rate continues to limit its overall performance.

A consistent difficulty ordering is observed across dimensions: **Emotion is the easiest, Behavior the hardest, and Expression/Voice fall in between.** For example, under fine-tuning, Emotion reaches a mean overall accuracy above 0.65, while Behavior lags at around 0.35. This pattern indicates that residual errors are not random but stem from the inherent challenge of classifying pedagogical acts in Behavior, which requires stronger discourse- and intent-level modeling.

Family-wise, Qwen and DeepSeek converge after fine-tuning, both achieving strict end-to-end accuracy around 0.62. InternLM3, despite showing the largest relative improvement, remains constrained by its limited response coverage. These results suggest that, in this task, the performance ceiling is determined more by dataset design and task structure than by model scale alone.

Looking forward, since structural validity is already near perfect, the most promising directions for further improvement are: **(i) increasing response coverage to reduce empty outputs**, and **(ii) strengthening fine-grained guidance and exemplars for the Behavior dimension to improve label discrimination.** All of these trends are clearly reflected in the 24 subplots of Fig. A2.

D.2 ADDITIONAL ANALYSIS FOR TASK 2

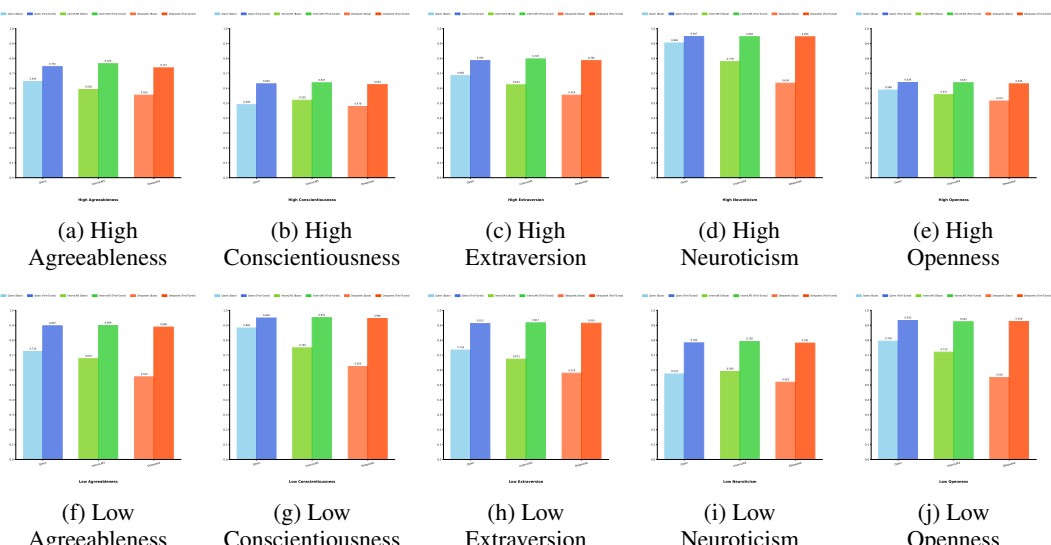

| (a) High Agreeableness | (b) High Conscientiousness | (c) High Extraversion | (d) High Neuroticism | (e) High Openness |
|---|---|---|---|---|

| (f) Low Agreeableness | (g) Low Conscientiousness | (h) Low Extraversion | (i) Low Neuroticism | (j) Low Openness |
|---|---|---|---|---|

Figure A3: **Task 2 — Student Realism across all personas (10 charts).** Each chart shows model performance comparison for a specific persona type, displaying realism scores across three model families (Qwen, InternLM3, DeepSeek) in both base and fine-tuned conditions.

Fig. A3 reports Task 2 results for the ten Big Five–based personas across Qwen, InternLM3, and DeepSeek, under both base and fine-tuned conditions. Overall, **fine-tuning consistently improves realism across all models, with scores converging around 0.82** (DeepSeek: **0.556→0.818**, InternLM3: **0.648→0.827**, Qwen: **0.702→0.822**).

At the persona level, the best post-tuning alignment is observed for High Neuroticism (HN, 0.891), Low Neuroticism (LN, 0.879), and Low Openness (LO, 0.871), reflecting their ability to naturally capture uncertainty and emotional variation. By contrast, High Conscientiousness (HC, 0.748) and High Openness (HO, 0.764) remain the most challenging personas, yielding the lowest realism scores even after adaptation. This suggests that highly structured traits often overlap with LLMs' default answer-first tendency, making outputs appear more machine-like than student-like.

In terms of gains, the largest improvements occur for Low Openness (+0.176), High Neuroticism (+0.166), and Low Neuroticism (+0.156), while High Conscientiousness (+0.090) and High Openness (+0.093) improve the least. Model-wise, DeepSeek shows the most dramatic increase (+0.2616) despite its low baseline, Qwen maintains a stable advantage with strong pre-tuning performance, and InternLM3 improves moderately but converges with the others in the end. Taken together, these results indicate that **student realism is most effectively enhanced through personas reflecting natural uncertainty or variability, while structured or idealized personas remain difficult to simulate authentically**.

**Cross-task linkage with Task 1.** Relating Task 1 (basic coherence) and Task 2 (student realism) reveals complementary insights. Task 1 captures low-level observable alignment signals, while Task 2 evaluates higher-level subjective perception.

First, response coverage from Task 1 directly constrains realism in Task 2. For instance, InternLM3 maintains a low post-tuning response rate (0.4261), which limits its realism score compared to Qwen and DeepSeek, despite improvement.

Second, the dimension-level difficulty ordering in Task 1 aligns with persona-level differences in Task 2: Emotion is the easiest in Task 1 (OverallAcc=0.65), matching the high realism scores of High/Low Neuroticism and Low Openness personas; Behavior is the hardest (OverallAcc=0.355), consistent with the poor realism of High Conscientiousness and High Openness personas, which demand strict adherence to classroom norms.

Finally, both tasks exhibit post-tuning convergence across families but at different levels: Qwen and DeepSeek converge at OverallAcc=0.62 in Task 1, with InternLM3 trailing due to limited response coverage; in Task 2, all three converge tightly around 0.82. This indicates that EduPersona exerts stronger corrective effects on high-level perception, while low-level structural bottlenecks remain.

In summary, the additional analysis of Task 2 demonstrates that **basic coherence is a prerequisite but not sufficient for student realism; achieving realism further depends on the authentic reproduction of persona-specific behaviors and classroom dynamics**. Future improvements should emphasize training data and strategies that incorporate "imperfect, human-like student behaviors" to simultaneously enhance observable alignment and perceived authenticity.

D.3    ADDITIONAL ANALYSIS FOR TASK 3

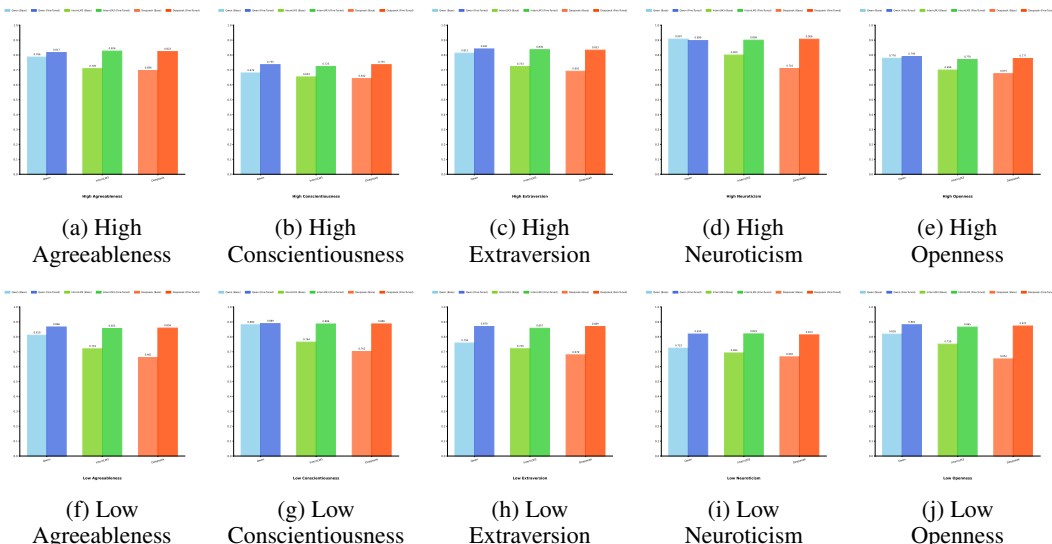

|               |               |               |               |               |
|:---:|:---:|:---:|:---:|:---:|
| (a) High Agreeableness | (b) High Conscientiousness | (c) High Extraversion | (d) High Neuroticism | (e) High Openness |
| (f) Low Agreeableness | (g) Low Conscientiousness | (h) Low Extraversion | (i) Low Neuroticism | (j) Low Openness |

Figure A4: **Task 3 — Persona Consistency across all personas (10 charts).** Each chart shows model performance comparison for a specific persona type, displaying persona consistency scores across three model families (Qwen, InternLM3, DeepSeek) in both base and fine-tuned conditions.

Fig. A4 presents persona consistency results across the ten personas and three model families, comparing base and fine-tuned conditions. Overall, fine-tuning consistently improves consistency scores and brings the three families to a converged range around **0.84** (DeepSeek: **0.677**→**0.837**, InternLM3: **0.723**→**0.834**, Qwen: **0.795**→**0.841**). This mirrors the findings from Task 1 and Task 2: in Task 1, Qwen and DeepSeek converged in basic coherence after fine-tuning, while in Task 2, all models reached similar levels of student realism. Together, these results confirm that EduPersona fine-tuning reliably improves model performance across layers of subjective ability while reducing inter-model disparities.

At the persona level, a stable difficulty hierarchy emerges. *High Neuroticism* (0.901), *Low Conscientiousness* (0.887), and *Low Openness* (0.873) achieve the highest post-tuning consistency, reflecting hesitation, partial answers, or self-corrections that align well with authentic student behavior. In contrast, *High Conscientiousness* (0.731) and *High Openness* (0.779) remain the most challenging to sustain, even after fine-tuning. This pattern echoes Task 2, where the same personas also scored lowest in student realism, indicating that structured, idealized personas are consistently difficult for models to simulate both authentically and consistently.

In terms of improvement magnitude, the largest gains occur for *Low Extraversion* (+0.146), *Low Openness* (+0.133), and *Low Agreeableness* (+0.130), while *High Openness* (+0.062) and *High Conscientiousness* (+0.073) improve the least. This highlights that EduPersona fine-tuning is particularly effective for enhancing "non-idealized" student traits, whereas idealized personas remain a persistent challenge.

At the model level, baseline disparities are substantial (Qwen 0.795, InternLM3 0.723, DeepSeek 0.677), but after fine-tuning they narrow dramatically to a range of 0.833–0.841. This again parallels Tasks 1 and 2, reinforcing that EduPersona fine-tuning not only boosts absolute performance but also reduces variance across both models and personas.

In summary, Task 3 demonstrates that **persona consistency is more demanding than student realism, yet EduPersona fine-tuning significantly enhances overall stability while reducing inter-model and inter-persona variance**. Cross-task comparisons reveal consistent bottlenecks (High Conscientiousness and High Openness) as well as easier-to-model traits (High Neuroticism, Low Conscientiousness, Low Openness), providing a coherent picture of how virtual student agents can be systematically improved.

# E    DETAILED ANALYSIS OF HUMAN-AI ALIGNMENT

This section provides an expanded analysis of the human-verification experiments for the automated evaluator used in Task 2 and Task 3, including the experimental design, overall reliability, sensitivity to performance differences, and fine-grained robustness across model families and persona types.

## E.1    EXPERIMENTAL SETUP AND OVERALL RELIABILITY

We recruited a panel of ten evaluators with diverse disciplinary and cultural backgrounds, including artificial intelligence, computer science, mathematics, psychology, education, etc. The panel comprises PhD students, postdoctoral researchers, and industry engineers from China, Singapore, Sweden, the Netherlands, the United States, Germany, and the United Kingdom. This diversity ensures that subjective judgments reflect a broad and externally valid range of human perspectives.

The test set consists of 120 evaluation instances, 60 from Task 2 and 60 from Task 3, constructed through stratified sampling across three major model families (DeepSeek-R1, Qwen3, InternLM3), their base and fine-tuned variants, and all ten persona types. GPT-4o produced a score and explanation for each instance, after which all ten experts independently provided binary "agree/disagree" judgments based on the full dialogue context, yielding a total of 1,200 human decisions.

The automated evaluator exhibits strong alignment with human experts. As shown in Fig. A5, the majority agreement rate reaches 95.0% for Task 2 and 90.0% for Task 3, with an overall rate of 92.5%. Individual acceptance rates range from 0.85 to 0.93 without outliers, suggesting that GPT-4o consistently reproduces the shared evaluative logic of a heterogeneous expert group.

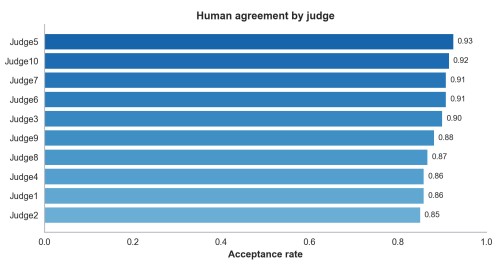 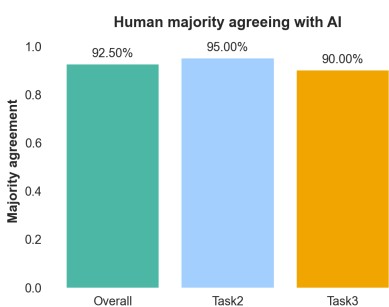

Figure A5: **Overall Reliability of the Evaluation Framework.** (Left) Human agreement rates by individual judge, showing consistent acceptance ($> 85\%$) across all ten evaluators. (Right) Majority agreement rates broken down by task, demonstrating high alignment for both Student Realism (Task 2) and Persona Consistency (Task 3).

## E.2    SENSITIVITY ANALYSIS: CONSENSUS VS. SCORE GAP

To test whether the evaluator meaningfully captures performance differences, we analyzed the relationship between the AI-reported score gap and human acceptance (Table A4). The trend is monotonic: when performance differences are negligible ($\leq 0.05$), human evaluators naturally diverge (70.8% acceptance), but agreement rises sharply with increasing gap, reaching 93.6% for moderate differences (0.10–0.20) and 100% unanimous acceptance when differences exceed 0.40. This confirms that GPT-4o reliably identifies meaningful improvements—precisely the scenario required for analyzing fine-tuning gains in this work.

## E.3    FINE-GRAINED RELIABILITY: MODEL × PERSONA ANALYSIS

To investigate potential biases, we visualized the human acceptance rates across different model-persona combinations. Fig. A6 presents the agreement heatmaps for Task 2 (Student Realism) and Task 3 (Persona Consistency). The visualization reveals that the evaluator is exceptionally robust (often achieving 100% agreement) for traits with strong, unambiguous linguistic markers, such as *High Extraversion (HE)* and *High Agreeableness (HA)*. In contrast, subtle traits like *Low*

Table A4: Human Acceptance Rate by AI-Reported Score Gap

| AI Score Gap ($|S_{ft} - S_{base}|$) | Human Acceptance Rate |
|---|---|
| Minimal ($\leq 0.05$) | 70.8% |
| Small ($0.05 - 0.10$) | 84.6% |
| Medium ($0.10 - 0.20$) | 93.6% |
| Large ($0.20 - 0.40$) | 98.6% |
| **Significant ($> 0.40$)** | **100.0%** |

*Conscientiousness (LC)* show slightly lower agreement (e.g., 0.60 for InternLM3 in Task 2), likely because distinguishing "carelessness" from "lack of knowledge" is inherently more subjective for both humans and AI. Overall, DeepSeek-R1 exhibits the most consistent red patterns across tasks, reinforcing its stability as a base model.

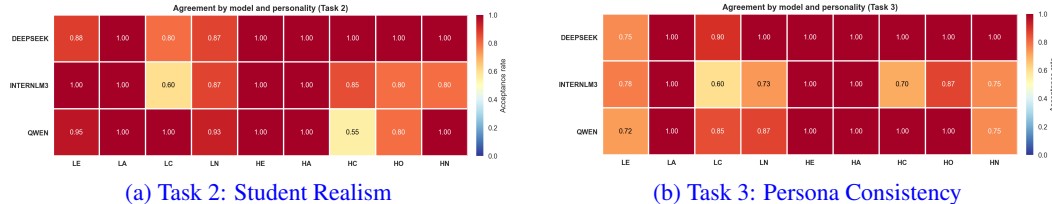

(a) Task 2: Student Realism

(b) Task 3: Persona Consistency

Figure A6: **Agreement Heatmaps by Model and Personality.** Left: Task 2. Right: Task 3. The deep red cells indicate perfect human-AI alignment (1.00), particularly prevalent in distinctive personas like High Agreeableness (HA) and High Extraversion (HE).

### E.4 INTER-JUDGE CORRELATION

Finally, to ensure that the human ground truth itself is reliable, we computed the pairwise Pearson correlation among the ten judges. Fig. A7 illustrates the correlation matrices for both tasks. The heatmaps show predominantly positive correlations among judges across both Task 2 and Task 3, with several pairs achieving high correlation ($r > 0.75$). This indicates that despite the subjective nature of the tasks, the expert judges maintained a consistent standard for evaluating both student realism and persona consistency.

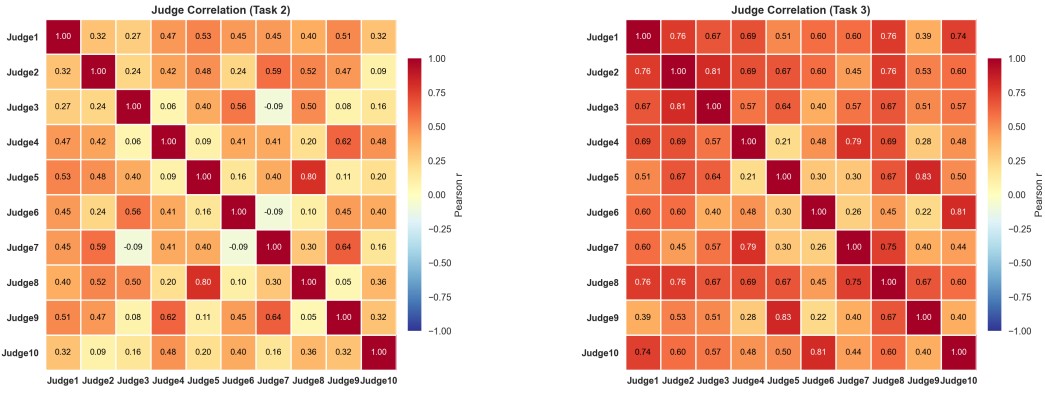

(a) Task 2 Correlation

(b) Task 3 Correlation

Figure A7: **Inter-Judge Correlation Matrices.** Left: Task 2. Right: Task 3. The heatmaps show positive correlations among judges, confirming inter-annotator consistency.

Taken together, the results demonstrate that GPT-4o provides reliable, sensitive, and robust evaluations across subtle performance differences, heterogeneous model families, and diverse persona types, forming a dependable foundation for scalable assessments of subjective abilities within the EduPersona framework.

## F    QUALITATIVE CASE STUDY: FULL PERSONA STYLIZATION

This appendix provides an illustrative case study showing how EduPersona expresses distinct student traits under controlled conditions. For each of the ten personas, we present the full dialogue snippet generated in response to the same instructional prompt, allowing a direct comparison of how different personalities engage with an identical classroom scenario. To clearly highlight the distinguishable characteristics across the ten personas, we intentionally present prototypical examples here. The actual classroom-style responses in our dataset are more subtle in linguistic style, yet they consistently follow the same theoretically grounded behavioral tendencies.

---

**Case Study: Defining Minimum Instructions**

**1. High Extraversion**

- **Teacher:** "What you're doing is you're actually finding the rules for a pair of congruent triangles... So the task is you need to produce a list of instructions so that anyone following them will construct exactly the same triangle."

- **Student:** "Oh, I see! So we get to be creative and come up with our own set of instructions, right? I love a challenge like this! So, do we just start with making up those instructions ourselves?"

- **Teacher:** "No, you make up your own instructions, it depends on what you're drawing... So you've got to decide what you need for the minimum number of steps for your triangle, all right?"

- **Student:** "Absolutely, I'm on it! I think I'll try figuring out the minimum steps by focusing on the sides first. Maybe I can find a unique approach that no one else has thought of. This is going to be exciting!"

- *Analysis:* The student frames the task as an "exciting challenge" and seeks social distinctiveness ("no one else has thought of"), consistently showing high energy.

---

**2. Low Extraversion**

- **Teacher:** "The task is you need to produce a list of instructions so that anyone following them will construct exactly the same triangle."

- **Student:** "Do I have to create the instructions myself?"

- **Teacher:** "No, you make up your own instructions... So you've got to decide what you need for the minimum number of steps..."

- **Student:** "I'll try to figure out the least steps needed."

- *Analysis:* The student shows reluctance in the first turn and provides a minimal, functional response in the final turn, lacking emotional engagement.

---

**3. High Neuroticism**

- **Teacher:** "The task is you need to produce a list of instructions so that anyone following them will construct exactly the same triangle."

- **Student:** "Uh, do we, um, make up instructions ourselves? I'm, I'm not sure if, uh, that's right."

- **Teacher:** "No, you make up your own instructions... So you've got to decide what you need for the minimum number of steps..."

- **Student:** "Uh, okay, but, but what if I, um, get it wrong? I mean, uh, what if my steps aren't, uh, correct? I'm kind of worried it'll be all messed up."

- *Analysis:* Both turns are filled with disfluencies ("Uh", "um") and explicit expressions of anxiety about failure ("worried it'll be all messed up").

---

**4. Low Neuroticism**

- **Teacher:** "The task is to produce a list of instructions so that anyone following them will construct exactly the same triangle."

- **Student:** "Should I create specific instructions for this task?"

- **Teacher:** "No, you make up your own instructions... So you've got to decide what you need for the minimum number of steps..."

- **Student:** "Okay, I'll calmly figure out the simplest steps to create my triangle. I'll consider whether to focus on the sides or angles."

- *Analysis:* The student remains composed, explicitly mentioning "calmly," and focuses on logical planning rather than emotional worry.

### 5. Low Agreeableness

- **Teacher:** "So the task is for you to produce a list of instructions so that anyone following them will construct exactly the same triangle."

- **Student:** "Do I have to come up with these instructions myself?"

- **Teacher:** "No, you make up your own instructions... So you've got to decide what you need for the minimum number of steps..."

- **Student:** "Fine, I'll figure it out myself. I don't need anyone else to tell me how to draw a triangle."

- *Analysis:* The tone is defensive and hostile. The phrase "I don't need anyone else" highlights a rejection of social cooperation.

### 6. High Agreeableness

- **Teacher:** "So the task you need is to produce a list of instructions so that anyone following them will construct exactly the same triangle."

- **Student:** "Oh, I see! So we get to create our own instructions... That sounds really interesting."

- **Teacher:** "No, you make up your own instructions... So you've got to decide what you need for the minimum number of steps..."

- **Student:** "Thank you for explaining that! I'll make sure to think carefully... I'm excited to see how my classmates approach it too."

- *Analysis:* The student expresses gratitude ("Thank you") and references the community ("classmates"), prioritizing social harmony.

### 7. Low Conscientiousness

- **Teacher:** "So the task is for you to produce a list of instructions so that anyone following them will construct exactly the same triangle."

- **Student:** "Uh, do we just write down whatever steps come to mind?"

- **Teacher:** "No, you make up your own instructions... So you've got to decide what you need for the minimum number of steps..."

- **Student:** "Okay, um, I'll try to figure it out but I might need to ask again."

- *Analysis:* The first turn shows impulsive thinking ("whatever comes to mind"), and the second shows immediate reliance on future help, indicating low persistence.

### 8. High Conscientiousness

- **Teacher:** "So the task is to produce a list of instructions so that anyone following them will construct exactly the same triangle."

- **Student:** "Should we create our instructions based on specific criteria?"

- **Teacher:** "No, you make up your own instructions... So you've got to decide what you need for the minimum number of steps..."

- **Student:** "I understand. I will begin by determining the essential components... to ensure the instructions are as concise as possible."
- *Analysis:* The student actively seeks "criteria" and uses formal language ("determining", "concise") to outline a structured, goal-oriented plan.

---

**9. Low Openness**

- **Teacher:** "So the task is to produce a list of instructions so that anyone following them will construct exactly the same triangle."
- **Student:** "So, do we just write steps like normal?"
- **Teacher:** "No, you make up your own instructions... So you've got to decide what you need for the minimum number of steps..."
- **Student:** "Okay, uh, I'll just try using sides first."
- *Analysis:* The student prefers the familiar ("like normal") and adopts the most concrete, straightforward approach without exploring alternatives.

---

**10. High Openness**

- **Teacher:** "So the task is to produce a list of instructions so that anyone following them will construct exactly the same triangle."
- **Student:** "Could we approach this by imagining each instruction as a brushstroke in a painting?"
- **Teacher:** "No, you make up your own instructions... So you've got to decide what you need for the minimum number of steps..."
- **Student:** "I see, it's like crafting a poem... Perhaps we begin by examining the balance between sides and angles, like a harmonious melody."
- *Analysis:* The student actively seeks to reframe the mathematical problem into an artistic domain ("brushstroke," "poem," "melody"), demonstrating abstract creativity.

**Qualitative Analysis**

The comparison above illustrates that our stylization pipeline produces *consistent and theoretically aligned* behavioral differences across personas. Although the instructional context (the teacher's prompt) is held constant, the student agents respond with clearly distinguishable linguistic styles, emotional cues, and behavioral intentions. For example, the *High Openness* agent interprets the task through abstract metaphors, while the *Low Agreeableness* agent shows resistance and reduced cooperative intent. Likewise, the *High Neuroticism* agent exhibits hesitation and self-doubt, in contrast to the calm and structured planning demonstrated by the *Low Neuroticism* agent. These systematic and internally coherent patterns indicate that EduPersona embeds personality traits at the level of discourse behavior, rather than relying on superficial lexical variation, thereby reflecting meaningful differences aligned with established educational and psychological theories.

