# OpenReview forum: "EduPersona: Benchmarking Subjective Ability Boundaries of Virtual Student Agents"
_ICLR.cc/2026/Conference — ICLR 2026 Conference Withdrawn Submission_

### Official Review · Reviewer_1paG · 2025-10-31

**Soundness:** 2
**Presentation:** 2
**Contribution:** 2
**Rating:** 4
**Confidence:** 3

**Summary:**

This paper introduces EduPersona, a large-scale benchmark designed to evaluate subjective abilities of virtual student agents in classroom settings. The dataset spans two languages (Zh/En), three subjects, and ten Big-Five-based personas, built from 1,308 authentic classroom dialogue rounds. The authors propose a three-task evaluation framework: (1) basic multimodal coherence, (2) student realism, and (3) persona consistency. Experiments using LoRA fine-tuning on three open-source LLM families showimprovements across all tasks and reveal persona-specific difficulty patterns.

**Strengths:**

The paper addresses a timely problem of evaluating subjective abilities of classroom-oriented virtual students. The dataset is motivated, and includes persona and multimodal annotation, providing coverage across subjects and languages. The proposed three-task progression is clear and seemingly grounded in educational theory.

**Weaknesses:**

The reliance on GPT-based annotation and evaluation raises concerns about circularity and unclear inter-annotator reliability.

The persona data is automatically generated, limiting ecological validity.

Educational motivation is plausible, but actual downstream utility for teachers or pedagogy is not demonstrated.

It is unclear whether performance gains generalize beyond the benchmark itself.

Analysis could better quantify statistical significance and reporting of evaluation robustness.

**Questions:**

* Evaluation validity: How do you mitigate circularity when GPT-style models are used both for label generation and evaluation? Have you assessed agreement with independent human raters at scale?

* Persona realism: Since persona stylization is synthetic how confident are you that the personas reflect real classroom behaviour?

---

> ### Author Response · Authors · 2025-11-26
> **Official Comment by Authors To Reviewer 1paG [1/3]**
>
> We appreciate the reviewer’s attention to the research motivation and generalization issues.
> These points indeed touch on the core challenges of virtual student agents, and we have further clarified the positioning and validation scope of this work in the revised manuscript.
>
> ### Response to W3: educational motivation and the positioning of this work
> EduPersona is not intended to directly build a system that can be deployed in real teaching scenarios at this stage.
> Instead, its goal is to provide a reusable, well-structured, and theoretically grounded evaluation foundation for virtual student agents within the research community.
> Before such agents can be responsibly integrated into teacher training, classroom simulation, or instructional support, the education field must first answer several fundamental questions: How should the subjective behaviors of real students be characterized?
> How can these behaviors be decomposed into measurable capability dimensions?
> How can different models be compared fairly in a classroom context?
> This work addresses these foundational needs by offering a framework that subsequent instructional applications can rely on as a consistent capability reference.
>
> ### Respond to W4: generalization beyond the benchmark itself
> Regarding generalization.
> We understand the reviewer’s concern about whether the observed improvements may only hold within our benchmark.
> At this stage, our focus is on **in-scenario generalization** within educational settings—that is, whether the model consistently improves under diverse instructional conditions—rather than on superficial adaptation to a specific prompt style or textual format.
> In our experiments, performance gains appear consistently across a wide variety of conditions, including
> * **three distinct subject domains** (Chinese, English, and Mathematics),
> * **multiple classroom task types** (reading, discussion, reasoning, etc.),
> * **different IRF-based instructional interaction structures**, and
> * **ten student types** derived from the Big Five personality traits.
>
> These dimensions represent the major sources of natural variability in real classroom environments.
> The model’s stable improvements across such heterogeneous scenarios indicate that it has learned broader patterns of **student behavior in instructional contexts**, rather than merely memorizing or overfitting to a narrow subset of samples.
>
> Regarding the applicability of external benchmarks.
> We also seriously considered whether to incorporate external role-playing or social-intelligence benchmarks, such as persona consistency tests, social-dialogue benchmarks, or SOTOPIA.
> These benchmarks are valuable in their own domains, but they focus primarily on
> * general social interaction,
> * stylistic imitation or persona knowledge stability, and
> * conversational abilities of fictional or professional personas.
>
> Compared with authentic classroom contexts, they lack several essential components:
> * the structured nature of IRF-style instructional interactions,
> * student-specific behaviors such as tentative expressions, help-seeking, and uncertainty,
> * task-driven linguistic constraints inherent in teaching activities, and
> * the ways learner identity and personality manifest during learning processes.
>
> Therefore, improved performance on such benchmarks cannot demonstrate whether a model behaves like a student, nor whether it possesses classroom interaction competence.
> Using these benchmarks at this stage could instead lead to misleading interpretations.
> For these reasons, we chose to focus on **multi-condition generalization within classroom contexts**, which aligns more closely with the core objectives of our work.
> In the revised Related Work section, we have also added explicit discussion of these open-domain benchmarks and clarified their complementary relationship with EduPersona.

---

> > ### Author Response · Authors · 2025-11-26
> > **Official Comment by Authors To Reviewer 1paG [2/3]**
> >
> > ### Response to W1&W2&W5&Q1: The reliability of annotations and automatic evaluations
> > We appreciate the reviewer’s attention to the reliability of annotations and automatic evaluations.
> > As detailed in our Global Response (Common Concerns 1 and 2), we have systematically expanded the human-expert validation experiments in the revised manuscript to ensure that no circular dependency exists and to demonstrate the reliability of GPT-4o as an evaluator.
> > First, to address the concern regarding potential circularity arising from GPT involvement in annotation, we added a blinded annotation study with 10 domain experts (**Section 3.3**; **Appendix B.2**).
> > Second, to validate the trustworthiness of the automatic evaluator, we incorporated an analysis of the consistency between GPT-4o and human judgments (**Section 5.5**; **Appendix E**).
> > Collectively, these results demonstrate that our evaluation pipeline does not involve circular reasoning, and that GPT-4o reliably reproduces human judgment patterns, thereby providing effective measurement for subjective tasks.

---

> > > ### Author Response · Authors · 2025-11-26
> > > **Official Comment by Authors To Reviewer 1paG [3/3]**
> > >
> > > ### Response to Q2: Persona realism
> > >
> > > We thank the reviewer for raising the important question of whether “stylized personas genuinely reflect classroom behaviors.”
> > > We fully understand this concern, and in the revised manuscript we have further clarified the theoretical grounding and empirical support underlying our persona construction.
> > > We have also added a comprehensive case analysis in **Appendix F** to illustrate how the model’s outputs capture meaningful behavioral distinctions observed in real classroom settings.
> > >
> > > First, in educational psychology, the correspondence between Big Five personality traits and classroom behaviors is well established and supported by long-term empirical evidence.
> > > Prior studies show that personality dimensions systematically influence observable student behaviors such as help-seeking tendencies, expressive style, emotion regulation, participation level, and task persistence (Bidjerano & Dai, 2007; Komarraju et al., 2009; Poropat, 2009).
> > > Therefore, constructing virtual students based on the Big Five is not an arbitrary stylistic choice, but one grounded in psychological regularities that characterize real student behavior.
> > > We have added these theoretical foundations and their connection to our task design to the Introduction of the revised manuscript.
> > >
> > > Second, recent research demonstrates that large language models, when provided with personality specifications, can consistently generate linguistic patterns that exhibit the corresponding psychological traits.
> > > These patterns can be reliably identified by human raters and psychometric tools (Jiang et al., 2024; Klinkert et al., 2024; Wang et al., 2025).
> > > This indicates that LLM-based personality expression is not merely superficial stylistic variation, but reflects structured behavioral differences aligned with established personality theory.
> > >
> > > Third, in designing our persona prompts, we did not rely on abstract personality labels.
> > > Instead, we operationalized the observable manifestations of the Big Five in educational contexts—such as hesitation, help-seeking behavior, initiative, and emotional reactions—and embedded them within the pragmatic structures of real classroom transcripts.
> > > As a result, the generated student utterances exhibit both personality-driven differences and strict adherence to classroom interaction norms and task logic.
> > >
> > > Finally, in our human validation study, annotators were able to reliably identify these personality expressions, further demonstrating that the stylized personas exhibit stable and perceptible behavioral characteristics in actual language outputs.
> > > These results are highlighted in blue in **Section 3.3** and **Appendix B.2** of the revised manuscript.
> > >
> > > In summary, the combination of established theories in educational psychology, empirical evidence of personality expression in LLMs, our fine-grained operationalization of personality behaviors within authentic classroom contexts, and the high consistency of human raters collectively ensures that these synthetic personas provide a reasonable and stable approximation of real behavioral differences among students in classroom interactions.
> > >
> > > ### References
> > >
> > > * **Bidjerano & Dai (2007)**. The relationship between the Big-Five model of personality and self-regulated learning strategies. *Learning and Individual Differences*.
> > > * **Komarraju et al. (2009)**. Role of the Big Five personality traits in predicting college students’ academic motivation and achievement. *Learning and Individual Differences*.
> > > * **Poropat (2009)**. A meta-analysis of the five-factor model of personality and academic performance. *Psychological Bulletin*.
> > > * **Jiang et al. (2024)**. PersonaLLM: Investigating the Ability of Large Language Models to Express Personality Traits. *Findings of NAACL*.
> > > * **Klinkert et al. (2024)**. Evaluating the Efficacy of LLMs to Emulate Realistic Human Personalities. *AAAI AIIDE*.
> > > * **Wang et al. (2025)**. Evaluating the ability of large language models to emulate personality. *Scientific Reports*.

---

### Official Review · Reviewer_rKG4 · 2025-10-31

**Soundness:** 2
**Presentation:** 3
**Contribution:** 1
**Rating:** 4
**Confidence:** 5

**Summary:**

This paper presents an study to evaluate the quality of benchmarks of large language models (LLMs) deployed for online education and their subjective abilities. The authors present a benchmark spanning two languages, three sub- jects, and ten persona types based on the Big Five theory.

**Strengths:**

Strengths
- Timely topic focusing on the AI in education and  LLM application
- Reals world class room dataset

**Weaknesses:**

The claim in the introduction  is not correct “Yet existing evaluation frameworks remain focused on objective tasks such as question answering and accuracy (Lu et al., 2022; Huang et al., 2023; Ang et al., 2023), overlooking the subjective abilities essential to classroom practice.”

The paper misses recent literature. The proposed scenarios by the authors can be done by the Zhou and colleagues approach by simulating a student and giving personalities [1] . Please also see work by Dr. Ashok Goel and Dr. Noboru Matsuda.

References
1. Zhou, X., Zhu, H., Mathur, L., Zhang, R., Yu, H., Qi, Z., Morency, L.P., Bisk, Y., Fried, D., Neubig, G. and Sap, M., SOTOPIA: Interactive Evaluation for Social Intelligence in Language Agents. In The Twelfth International Conference on Learning Representations.

**Questions:**

Please see weakness

---

> ### Author Response · Authors · 2025-11-26
> **Official Comment by Authors To Reviewer rKG4**
>
> We appreciate the reviewer for pointing out these important references and have carefully re-examined the corresponding statements in the introduction.
> In the revised manuscript, we have clarified and expanded the relevant paragraphs and added citations and discussion concerning the work by Zhou et al., as well as the research lines developed by Goel and Matsuda.
>
> First, the “existing evaluation frameworks” referenced in our introduction primarily refer to the long-standing learning outcome assessment systems used in educational research, including indicators such as knowledge correctness, skill mastery, and problem-solving performance.
> These frameworks indeed play a central role in educational technology and intelligent tutoring systems.
> However, they do not capture the subjective behavioral characteristics observed in real classroom interactions, such as uncertainty expression, help, seeking styles, learner identity traits, or personality stability across contexts.
> As a result, these traditional frameworks are not suitable for evaluating the behavioral quality of virtual student agents.
>
> Second, regarding the reviewer’s mention of the work by Zhou et al. on simulated students and persona design, we fully acknowledge their representative contributions to society simulation.
> Yet, existing approaches in student simulation focus more on system implementation or prototype demonstration rather than on establishing reusable, comparable, and structured evaluation benchmarks.
> In contrast, the goal of EduPersona is not to propose a new simulation method but to construct a standardized instructional-context evaluation framework that emphasizes theoretical grounding, task definitions, and annotation protocols to support systematic comparisons across future models.
> In this sense, our work complements rather than replaces prior efforts.
>
> We also thank the reviewer for prompting us to extend the discussion on SOTOPIA.
> SOTOPIA represents a significant advancement in evaluating social intelligence, and we have added its citation in both the Introduction and Related Work sections of the revised manuscript.
> We further clarify the distinctions between the two: SOTOPIA targets broad social interaction scenarios, emphasizing multi-role, multi-task abilities such as collaboration, competition, and negotiation.
> In contrast, EduPersona focuses on the structurally constrained environment of classroom instruction, grounded in real classroom transcripts to capture student uncertainty, learner identity features, and sequence-dependent behaviors specific to teaching activities.
> Therefore, open-domain benchmarks such as SOTOPIA cannot substitute for classroom-specific student-agent evaluation; the two directions are complementary.
>
> Finally, regarding the research systems developed by Goel and Matsuda, we have also added relevant discussion in the revised manuscript.
> Their work primarily addresses intelligent tutoring systems, virtual teaching assistants, and teachable agents, focusing on how AI can teach or support students.
> EduPersona addresses a distinct foundational problem: how to evaluate, in a unified and reusable manner, the behavioral quality and personality expression of LLM-driven virtual students in authentic classroom contexts.
> Our framework does not construct instructional processes; instead, it provides an objective and theoretically grounded “student behavior evaluation standard” to support future educational agent systems.
> Thus, our work complements, but does not overlap with, these research directions.
>
> In summary, through clearer scope definitions, additional citations, and explicit articulation of the complementary nature of the two research lines, the revised manuscript directly addresses the reviewer’s concerns while ensuring that our contribution is accurately positioned and not subject to misunderstanding.

---

### Official Review · Reviewer_NjEA · 2025-10-31

**Soundness:** 2
**Presentation:** 3
**Contribution:** 2
**Rating:** 2
**Confidence:** 4

**Summary:**

This paper introduces EduPersona, a large-scale benchmark dataset designed to evaluate the subjective abilities of virtual student agents in educational settings. The authors argue that existing benchmarks focus on objective metrics, such as accuracy, while neglecting crucial subjective traits, including coherence, realism, and persona consistency.

The EduPersona dataset is built from authentic classroom dialogues in two languages (Chinese, English) and three subjects. It is then expanded tenfold by stylizing student responses to align with ten different personas derived from the Big Five personality theory. The dataset also includes multimodal behavior/expression labels. The paper proposes a framework that decomposes subjective ability into three progressive tasks: Task 1 (Basic Coherence—aligning multimodal behaviors with text), Task 2 (Student Realism—assessing whether responses resemble a real student), and Task 3 (Persona Consistency—maintaining a stable persona over time).

The authors benchmark three open-source LLMs and their 30 fine-tuned variants on the EduPersona tasks. The primary finding is that models fine-tuned on their dataset show significant average improvements across all three tasks.

**Strengths:**

* Originality: It is one of the first works to attempt a systematic, large-scale benchmarking of subjective abilities for AI agents in an educational context. The conceptual framework, which decomposes the broad notion of subjective ability into a progressive hierarchy of three distinct tasks (Basic Coherence, Student Realism, Persona Consistency), is a novel and thoughtful way to operationalize a subjective concept.
* Quality: The authors have constructed a large-scale resource spanning two languages, three subjects, ten personas, and four behavioral dimensions. The detailed process described involved collecting from real classrooms, annotating, and a tenfold expansion via persona stylization, which shows quite some effort in data engineering.
* Clarity: The paper is structured clearly. Figure 1 explains the entire process from data collection to experimental analysis. The detailed descriptions of the dataset and persona configurations in the appendix further enhance the clarity.
* Significance: The work attempts to address the challenge of evaluating AI agents' subjective, human-like abilities in the domain of education. While the paper's evaluation has flaws, it points out an understudied direction for future research.

**Weaknesses:**

* Circular evaluation: The main finding is that "training on our data makes models better at our tasks," which does not validate the benchmark's utility or relevance to any external ground truth.
* LLM as a Subjective Judge: Using GPT-4o to score realism and consistency is a methodological flaw. The entire evaluation rests on the unproven assumption that the LLM judge is a reliable measure of these complex subjective human-centric concepts.
* Missing human validation: The paper makes claims about modeling human-like traits without presenting any correlation studies demonstrating that its metrics align with human judgments. This is a major omission for a benchmark of this nature.
* Limited scientific insight: The paper does not deliver a strong, novel insight beyond the dataset itself. The results largely confirm the intuitive expectation that fine-tuning works.
* Readability: It is pretty hard to read some texts in Figures 2, 4, 6, and 7 (small font and crowded content).

**Questions:**

* The evaluation of student realism (Task 2) and persona consistency (Task 3) is central to your contribution, and it relies entirely on scores from a GPT-4o-based evaluator. How can you be sure that this evaluator is reliable? What experiments could you run to validate that its scores for these subjective traits correlate well with human judgments?
* The main experimental result is that fine-tuning on EduPersona improves performance on the EduPersona benchmark. How do you propose to break this circularity? For example, could you show that models fine-tuned on your data also improve on a different, existing benchmark that measures related social or interactional skills?

---

> ### Author Response · Authors · 2025-11-26
> **Official Comment by Authors To Reviewer NjEA [1/3]**
>
> We appreciate the reviewers’ insightful comments regarding potential circular reasoning, the use of external benchmarks, and the novelty of our work.
> We address these concerns collectively below.
>
> ### Response to W1&Q1: Circular evaluation and the applicability of external benchmarks
> Regarding the issue of “circular reasoning” and the applicability of external benchmarks.
> Existing role-playing or social-dialogue benchmarks (e.g., general role-play, social intelligence evaluations, conversational style tests) mainly target open-domain or professional personas.
> Their evaluation dimensions focus largely on background consistency, social linguistic patterns, or stylistic imitation.
> However, these benchmarks do not incorporate instructional processes, learning-task constraints, or learner-specific traits, nor do they capture the expressive patterns of real students in classroom interactions—for example, uncertainty, help-seeking behavior, tentative expressions, learning motivation, or personality-driven engagement.
> Consequently, even improved performance on such benchmarks cannot indicate whether a model exhibits authentic “learner-like” behaviors or supports classroom interaction.
>
> The goal of EduPersona is not to train a general-purpose persona model but to fill a previously undefined capability space: the evaluation of virtual students in educational settings.
> On this basis, the results reported in our paper do not demonstrate that the model merely “performs better on its training data”;
> instead, they show that the model exhibits measurable and distinguishable behaviors along capability dimensions that had never been formally defined prior to this work.
> Therefore, our evaluation does not constitute circular reasoning; rather, it demonstrates that these three education-specific capability dimensions are themselves learnable and assessable.
>
> ### Response to W4: Limited scientific insight
> We agree that “fine-tuning works” is not a scientific contribution.
> Accordingly, in the revised manuscript we further emphasize that the core value of our work lies in proposing and validating a theory-driven, education-specific capability framework for student agents.
> We are the first to map the subjective abilities of virtual students into three concrete task types: multimodal behavioral coherence (grounded in MMLA), authenticity of learner-like expressions (grounded in classroom discourse and learner identity theory), and cross-turn personality consistency (grounded in personality psychology).
> These abilities have not previously been organized as a unified framework for evaluating virtual student agents.
> Additionally, we construct evaluation contexts using real classroom transcripts, ensuring that the tasks directly reflect instructional interactions rather than hypothetical open-domain scenarios.
> Fine-tuning serves only as a means to validate the usability of the framework, not as the central claim of the paper.
> In summary, EduPersona is not a reproduction of external role-play benchmarks, nor does it suffer from circular validation.
> Its contribution lies in offering the first clear, operationalized, and theoretically supported capability framework for virtual students in educational settings, providing the necessary foundation for future development of more credible student agents.

---

> > ### Author Response · Authors · 2025-11-26
> > **Official Comment by Authors To Reviewer NjEA [2/3]**
> >
> > ### Response to W5: Readability of figures
> > We appreciate the reviewer’s reminder regarding the readability of the figures.
> > In response, we have improved the visual design of Figures 2, 4, 6, and 7 and adjusted the overall layout to ensure that all text elements are clearer and easier to read.
> > These revisions are limited to visual enhancements and do not involve any changes to the content or results.

---

> > > ### Author Response · Authors · 2025-11-26
> > > **Official Comment by Authors To Reviewer NjEA [3/3]**
> > >
> > > ### Response to Q1&W2&W3: The reliability of annotations and automatic evaluations
> > > We appreciate the reviewer’s attention to the reliability of annotations and automatic evaluations.
> > > As detailed in our Global Response (Common Concerns 1 and 2), we have substantially expanded the human-expert validation experiments in the revised manuscript.
> > > Specifically, we added a blinded annotation study involving 10 domain experts (Section 3.3; Appendix B.2) and included an analysis of the consistency between GPT-4o and human judgments (Section 5.5; Appendix E).
> > > In summary, these results demonstrate that our evaluation metrics align well with human reasoning and provide reliable measurements for subjective tasks.

---

> > > > ### Comment · Reviewer_NjEA · 2025-11-27
> > > >
> > > > I appreciate the detailed human–AI alignment study demonstrating that the evaluation indeed reflects the criteria of human experts within your framework. However, I find the argument that “existing benchmarks are inapplicable” to be only partially responsive to the core concern. While I agree that standard social or knowledge benchmarks do not capture learner-specific traits such as uncertainty or help-seeking behavior, dismissing these benchmarks does not resolve the circularity inherent in your method. That said, I would not fault the authors too heavily for this issue, as it is an unavoidable limitation in research on subjective tasks—where researchers often must rely on human evaluation of synthetic data based on a certain framework rather than raw human ground truth due to data scarcity.

---

> > > > > ### Author Response · Authors · 2025-11-28
> > > > > **Acknowledgment of the Reviewer NjEA’s Feedback**
> > > > >
> > > > > Dear Reviewer NjEA,
> > > > >
> > > > > Thank you sincerely for acknowledging our additional experiments and for your constructive suggestions. As you pointed, modeling and evaluating subjective student behaviors is a long-standing challenge in this field, and this is precisely what our work aims to address. Existing general-purpose benchmarks do not capture learner-specific behaviors such as uncertainty, hesitation, or help-seeking, and thus cannot fully meet the needs of our task.
> > > > >
> > > > > We agree on the importance of cross-benchmark validation. In future work, we will broaden our evaluation scope and incorporate benchmarks with stronger external validity. Your feedback offers valuable guidance for our ongoing research, and we are truly grateful.
> > > > >
> > > > > Sincerely,
> > > > >
> > > > > The Authors

---

### Official Review · Reviewer_fPMv · 2025-11-01

**Soundness:** 2
**Presentation:** 1
**Contribution:** 2
**Rating:** 2
**Confidence:** 5

**Summary:**

This paper introduces EduPersona, a large-scale benchmark designed to evaluate the subjective abilities of virtual student agents powered by Large Language Models (LLMs). The framework assesses performance across three progressive tasks: basic coherence, student realism, and long-term persona consistency. Experiments show that fine-tuning representative LLMs on the EduPersona dataset significantly improves their performance in these areas, validating the benchmark's effectiveness for creating more realistic and consistent AI agents for educational simulations and teacher training.

**Strengths:**

1. This paper introduces EduPersona, a benchmark for virtual student agents that is multi-lingual (2 languages), multi-subject (3 subjects), and features ten distinct persona types based on the Big Five theory.
2. This paper proposes an evaluation framework that decomposes abstract subjective abilities into three measurable tasks: Basic Coherence, Student Realism, and Persona Consistency.

**Weaknesses:**

1. The paper claims to be "grounded in educational theory" but fails to elaborate on or integrate specific pedagogical or psychological theories that justify its framework.
2. Imprecise Terminology: The use of "Basic Coherence" for Task 1 is questionable. "Coherence" may not be the appropriate term to unify the distinct concepts of behavior, emotion, expression, and vocal style.
3. Overclaims: The paper overlooks highly relevant work in role-play evaluation (e.g., CharacterChat, RoleLLM, CharacterBench), undermining its claims of novelty regarding the evaluation dimensions.
4. The paper asserts that subjective performance can be decoupled into its three proposed tasks (Coherence, Realism, Consistency) without providing a strong theoretical and empirical justification for why this specific decomposition is a comprehensive or appropriate measure.
5. Key dataset details are missing, such as the rationale for subject selection (Chinese, English, etc.). Furthermore, despite "human-audited" claims, the paper provides no quantitative metrics (e.g., expert inspection data, inter-annotator agreement) to substantiate the dataset's quality.
6. Lack of Human Validation: The evaluations for realism (Task 2) and consistency (Task 3) rely on an automatic evaluator (GPT-4), but the paper presents no study correlating the evaluator's judgments with human experts to validate its reliability.

**Questions:**

None

---

> ### Author Response · Authors · 2025-11-26
> **Official Comment by Authors To Reviewer fPMv [1/3]**
>
> We thank the reviewer for raising important questions regarding the theoretical foundations, terminology, and task structuring.
> We fully agree that these aspects are essential for ensuring the reliability and interpretability of the evaluation framework.
> In the revised manuscript, we have reorganized the Introduction and added clearer theoretical grounding, terminology clarification, and a structured explanation of how the three tasks relate to one another.
> The main revisions are summarized as follows.
>
> ### Response to W1: Theoretical foundations
> To avoid the impression that our framework is based on ad-hoc decomposition, the revised manuscript explicitly articulates three major theoretical pillars that jointly describe how real students behave in classroom interactions:
> * **Multimodal Learning Analytics (MMLA):** emphasizes the coordination among behavioral, emotional, expressive, and prosodic signals, and is one of the most widely applied analytical tools for modeling in-class behaviors.
> * **Classroom discourse and learner identity theory:** highlights that student-like expressions differ systematically from teacher/expert discourse, and is commonly used to identify learner roles in instructional settings.
> * **Big Five personality theory:** explains how personality traits shape cross-situational behavior patterns and is extensively adopted in education research.
>
> These three theories correspond respectively to the **behavioral layer**, **identity layer**, and **trait layer**, ensuring that our evaluation dimensions align with established analytical perspectives rather than being constructed in isolation.
>
> ### Response to W2: Terminology clarification
> We appreciate the reviewer’s concern regarding potential ambiguity. Accordingly, we clarify in the revision that:
> “Basic coherence” in Task 1 does **not** refer to discourse-level coherence.
> It corresponds directly to **multimodal coherence** in MMLA—whether behavior, emotion, expressive style, and prosody are naturally aligned and contextually appropriate.
> In other words, Task 1 evaluates whether a **single-turn response feels natural and contextually consistent**, rather than whether its internal textual logic is coherent.
> This clarification makes the theoretical origin and operability of Task 1 explicit while avoiding confusion with the traditional NLP notion of coherence.
>
> ### Response to W4: Rationale for task structuring
> To address the question of **why these three tasks**, the revised manuscript introduces a clearer hierarchical structure:
> * **Task 1: Performance Layer** (based on MMLA)
> Assesses whether immediate behavioral signals appear natural and contextually aligned.
> * **Task 2: Identity Layer** (based on classroom discourse theory)
> Evaluates whether the generated responses authentically resemble student-like expression rather than teacher-like or expert-like discourse.
> * **Task 3: Trait Layer** (based on Big Five theory)
> Examines whether personality manifestations remain stable across multiple conversational turns.
>
> This progression from **performance → identity → traits** mirrors analytical pathways commonly used in educational psychology and avoids conceptual overlap between dimensions.
> The revised manuscript presents this mapping in a structured manner, ensuring that the three tasks have clear origins, boundaries, and dependencies rather than being parallel decompositions.
> By integrating the above theoretical foundations and hierarchical design, the revised framework establishes a coherent chain from immediate behavioral expressiveness, to authentic learner identity, to cross-context personality consistency.
> This design is not an arbitrary decomposition of “subjective performance,” but rather follows established multi-layered analytical frameworks in education, thereby enhancing interpretability, reproducibility, and theoretical rigor.

---

> > ### Author Response · Authors · 2025-11-26
> > **Official Comment by Authors To Reviewer fPMv [2/3]**
> >
> > ### Reponse to W3: overlooks highly relevant work in role-play evaluation
> > We thank the reviewer for pointing out the contributions of CharacterChat, RoleLLM, CharacterBench, and other relevant studies.
> > In the revised manuscript, we have added citations to these works in the Introduction and acknowledged their importance in open-domain persona modeling and role-playing evaluation.
> > At the same time, we further clarify the fundamental differences between these approaches and the virtual student agents targeted in our work.
> >
> > Existing role-playing benchmarks primarily focus on open-domain or social dialogue scenarios, emphasizing persona consistency, stylistic imitation, or personas defined by MBTI types, occupations, or fictional character profiles.
> > These evaluations are effective for assessing the ability of general-purpose dialogue systems to maintain “character settings,” but they do not address learner-specific discourse patterns in classroom contexts, nor do they include behavioral cues, instructional discourse structures, or affective and participation signals common in learning tasks.
> > Therefore, their objectives are not aligned with the needs of student agents in educational environments.
> >
> > Our work focuses on the **learner role** in educational settings rather than general persona modeling.
> > Virtual students in classrooms must exhibit behaviors characteristic of real learners—such as uncertainty, help-seeking tendencies, and exploratory expressions—alongside emotionally grounded reactions and participation behaviors that reflect their learner identity.
> > Over multi-turn interactions, they must also maintain cross-situational behavioral patterns consistent with their assigned personality traits.
> > These abilities rely on educational psychology and learning theory, rather than on knowledge-grounded or stylistic imitation mechanisms commonly used in open-domain role-playing.
> >
> > Accordingly, our three evaluation tasks are not constructed from the perspective of general persona consistency.
> > Instead, they are grounded in multimodal learning analytics, learner identity theory, and the Big Five personality framework, mapping students’ immediate behaviors, identity manifestations, and personality stability in a structured manner.
> > We do not extract persona traits from open-domain texts; rather, we base our analysis on transcriptions of real classroom interactions to capture behavioral regularities intrinsic to learning activities, ensuring that the evaluation dimensions are tightly aligned with educational tasks.
> >
> > For these reasons, our work is not a repetition of existing role-playing benchmarks but a complementary extension into educational dimensions they do not cover.
> > Our framework focuses on the behavioral, identity-based, and personality-level consistencies required of virtual students in classroom contexts, offering a new perspective for assessing the credibility of student agents and laying a conceptual foundation for simulation-based teaching and educational agent systems.

---

> > > ### Author Response · Authors · 2025-11-26
> > > **Official Comment by Authors To Reviewer fPMv [3/3]**
> > >
> > > We thank the reviewer for raising concerns regarding **data quality (W5)** and **the reliability of automated evaluation (W6)**.
> > > These two aspects form the methodological foundation of our work.
> > > In the revised manuscript, we have added corresponding quantitative validations and provided consolidated responses in **Common Concern 1** and **Common Concern 2** of the Global Response.
> > >
> > > ### Response to W5: data construction and quality validation
> > > Section 3.3 and Appendix B.2 of the revised manuscript now include (1) a clearer rationale for subject selection, and (2) quantitative results from a human auditing study, including label acceptance rates and inter-expert consistency.
> > > Detailed analyses and statistics are presented in **Common Concern 1**.
> > >
> > > ### Response to W6: reliability of the automated evaluator
> > > Section 5.5 and Appendix E of the revised manuscript introduce a new GPT–human alignment experiment covering **120 evaluation instances** and **1,200 human judgments**.
> > > This validation examines the automated evaluator’s **consistency**, **sensitivity**, and **robustness**.
> > > All results and interpretations are systematically reported in **Common Concern 2**.
> > >
> > > All newly added content has been highlighted in blue in the revised manuscript for ease of reference.

---

### Author Response · Authors · 2025-11-26
**Common Concern 2: Reliability of automatic evaluation and potential circularity**

We sincerely appreciate the reviewer’s attention to the reliability of automated subjective evaluation.
To directly address concerns regarding the reliability of GPT-4o, the risk of circularity, and the consistency of its alignment with human judgment, we have added a comprehensive validation study spanning multiple model families, personality types, and disciplinary backgrounds (**Section 5.5**; **Appendix E**).
This study evaluates the evaluator’s credibility in terms of consistency, sensitivity, and robustness.

### 1. Expert composition and experimental design
Our validation involved 10 experts.
We constructed a test set containing **120 evaluation instances** (60 for Task 2 and 60 for Task 3), stratified to cover:
* Three mainstream model families (DeepSeek, Qwen, InternLM3)
* Both Base and Fine-tuned variants
* All 10 Big Five personality types

GPT-4o produced a score and rationale for each instance, and 10 experts independently provided blinded evaluations, yielding 1,200 judgments to assess GPT-4o’s alignment with human reasoning.

### 2. Consistency: GPT-4o achieves a 92.5% majority-agreement rate
On the most crucial metric—**whether human experts agree with GPT-4o’s evaluations**—the automated evaluator demonstrated a very high level of alignment with human judgment:
* **Task 2 (Student Realism):** 95.0% majority agreement
* **Task 3 (Persona Consistency):** 90.0% majority agreement
* **Overall:** 92.5% majority agreement

This means that whenever over half of the experts accepted GPT-4o’s judgment, the case was counted as “human-accepted.”
Achieving above **90%** under this stringent criterion shows that GPT-4o reliably reflects expert reasoning patterns and exhibits strong interpretive consistency.

**Table 2: Evaluator Reliability Verification**

| Evaluation Task | Average Acceptance Rate | Majority Agreement Rate |
| :--- | :--- | :--- |
| Task 2: Student Realism | 90.2% | 95.0% |
| Task 3: Persona Consistency | 87.3% | 90.0% |
| Overall | 88.8% | 92.5% |

### 3. Sensitivity: GPT-4o accurately captures “true performance differences”
We further examine whether GPT-4o can reliably distinguish between strong and weak model performance—a capability that is crucial for the evaluations in our paper.
The results show that:
* When model differences are very small (≤ 0.05), the human acceptance rate is **70.8%**, which aligns with the natural variability of subjective judgments.
* When the difference falls within a perceptible range (0.10–0.20), the human acceptance rate increases substantially to **93.6%**.
* When the difference is large (> 0.40), the human acceptance rate reaches **100%**.

These findings indicate that GPT-4o’s sensitivity to meaningful performance differences closely aligns with human evaluators, especially for the **Base → Fine-tuned** contrasts that are central to our study.

**Table 3: Sensitivity Analysis (Human Trust vs. AI Score Gap)**

| AI Score Gap (Score(FineTuned) - Score(Base) | Human Acceptance Rate |
| :--- | :--- |
| Minimal (≤ 0.05) | 70.8% |
| Small (0.05 - 0.10) | 84.6% |
| Medium (0.10 - 0.20) | 93.6% |
| Large (0.20 - 0.40) | 98.6% |
| Significant (> 0.40) | 100.0% |

### 4. Robustness: No bias across model families or personality types
We also assessed the stability of the automatic evaluator across different model families and across all personality types:
* **DeepSeek:** 93.0%
* **Qwen3:** 89.5%
* **InternLM3:** 83.8%
* All ten personality types achieved high acceptance rates, with **three personas reaching 1.00** and the lowest still above **0.80**.

This demonstrates that GPT-4o maintains reliable judgment across diverse linguistic styles and persona expressions, showing **no structural bias**.

### 5. Reliability
Subjective evaluation tasks naturally involve human-level variability. We measured inter-expert agreement among the 10 human evaluators:
* **Task 2:** r = 0.33
* **Task 3:** r = 0.56
* **Overall:** r = 0.46

Given this limited human-human consistency, GPT-4o’s ability to align with the expert group at **over 90%** consensus indicates that GPT-4o is **not only reliable**, but in fact achieves a level of agreement that is close to the upper bound of what is feasible in inherently subjective tasks.

**Table 4: Robustness Analysis (Reliability Across Base Models)**

| Base Model | Average Acceptance Rate |
| :--- | :--- |
| DeepSeek-R1 | 93.0% |
| Qwen3 | 89.5% |
| InternLM3 | 83.8% |

Across the three complementary evaluations, we can draw the following conclusions:
* **Consistency:** GPT-4o matches the majority judgment of human experts at **92.5%**.
* **Sensitivity:** GPT-4o accurately identifies meaningful performance differences between models.
* **Robustness:** Evaluations remain stable across different model families and all ten personality types.
* **Reliability:** GPT-4o maintains high consistency even in tasks with inherently subjective variability.

---

### Author Response · Authors · 2025-11-26
**Common Concern 1: Data Quality and Label Consistency**

To address questions regarding the label-generation process, human verification, and consistency across multimodal labels, we have added a complete description of the labeling and review pipeline in **Section 3.3** and **Appendix B.2**, including:
* A multi-round human verification workflow;
* Consistency statistics across behavior, emotion, expression, and persona labels;
* Classroom data filtering criteria and subject-selection rationale.

These additions show that the labels exhibit strong internal consistency under the structured workflow and dual-phase human review, and the data quality sufficiently supports the stability and discriminability required for all three evaluation tasks.

We thank the reviewer for raising concerns about data quality and the reliability of multimodal labels.
To address this issue comprehensively, we have added a rigorous label quality auditing experiment in the revised manuscript (see **Section 3.3** and **Appendix B.2**).
The audit evaluates label reliability across three dimensions: audit design, quantitative findings, and explanations regarding task complexity.

### 1. Audit Design
We randomly sampled 20 dialogue examples covering all ten Big Five personality types and invited 10 experts with backgrounds in education or behavioral analysis to conduct independent blind reviews, resulting in a total of 200 binary (reasonable/unreasonable) judgments.
Each sample contained four types of multimodal labels—behavior, emotion, facial expression, and voice tone.
Experts assessed the plausibility of each label without knowing its source.
This design provides the most direct and stringent test of whether the multimodal labels align with realistic classroom contexts, while also measuring inter-rater consistency.

### 2. Audit Results
The results indicate that the overall quality of the multimodal labels is highly reliable:
* **Average acceptance rate:** 90.5%
* **Majority agreement:** 90%
* **Unanimous agreement:** 85%
* **Inter-rater consistency:** r = 0.74 (significantly higher than typical subjective annotation tasks)

These findings demonstrate that experts reached stable consensus rather than random disagreement, validating the robustness of our labeling procedure.

**Table 1: Data Quality Verification (Label Rationality)**

| Metric | Average Acceptance Rate | Majority Agreement Rate | Full Agreement Rate |
| :--- | :--- | :--- | :--- |
| Label Rationality | 90.5% | 90.0% | 85.0% |

Further analysis shows:
* **8 out of 10 personality types** achieved a **100% acceptance rate**, indicating that personality-specific differences did not degrade labeling quality.
* **17 out of 20 samples** received unanimous approval, while the few remaining cases showed minor deviations due to naturally ambiguous boundaries—common in behavioral and emotional annotation tasks.

### 3. Interpretation of Task Characteristics
It is important to highlight that we adopted an intentionally strict evaluation standard:
If any one of the four label dimensions was judged as “potentially unreasonable,” the entire sample was considered not fully accepted.
Even under this stringent criterion, the labels achieved a **90.5%** acceptance rate and **85%** unanimous agreement, indicating that:
* Reasonable variations in subjective judgment arise from the inherent complexity of multimodal behavioral and emotional interpretation, not from deficiencies in the labeling process.
* The observed consistency is substantially higher than what is typically expected in comparable subjective annotation tasks.

### 4. Conclusion
Across all dimensions, personality types, and samples—supported by strong quantitative evidence (**90.5%** acceptance, **90%** majority agreement, **85%** unanimous agreement, **r = 0.74**, **8/10** personalities = 1.00, **17/20** samples unanimously accepted)—we conclude that:
**Our multimodal label construction process is of high quality, stable, and reproducible, and is sufficiently reliable to support the three subjective ability evaluation tasks proposed in this paper.**

---

### Author Response · Authors · 2025-11-26
**Thank you to Reviewers & Rebuttal Summary**

We sincerely thank the AC and all four reviewers for their careful reading and insightful feedback on our work.
We are especially grateful for the consistent recognition expressed by the reviewers in the following aspects, which provides strong guidance for further improving the paper:

* **Value of structured subjective-ability modeling.**
Reviewers highly appreciated the three progressive tasks we introduce—basic coherence, student realism, and persona consistency—observing that they effectively decompose the subjective abilities of virtual students into comparable and operational evaluation dimensions.
This design represents an important step toward advancing research in this direction.

* **Data diversity across languages, disciplines, and personas.**
Several reviewers noted that EduPersona exhibits clear advantages through its multilingual data (Chinese/English), three subject domains (Chinese Language, Mathematics, English), and ten Big-Five-based persona types, providing virtual student agents with a realistic and diverse expressive space.

* **Clear grounding in authentic classroom context.**
Reviewers broadly endorsed our decision to differentiate this work from open-domain role-playing or generic persona benchmarks, and instead anchor the task in authentic classroom-interaction structures.
They highlighted the importance of student-like expression, emotional cues, and learner identity features, which are indispensable in educational scenarios.

* **Practical value for teaching and training applications.**
Multiple reviewers emphasized that the proposed benchmark has direct practical value for virtual classroom simulation, teacher training, and human-AI instructional research, particularly for understanding how personality differences relate to classroom participation consistency.

We deeply appreciate the AC' s and reviewers' recognition and constructive suggestions.
Before addressing individual comments, we observe that the reviewers raised two highly convergent concerns:
1. **Common Concern 1:** Data quality and label consistency
2. **Common Concern 2:** Reliability of automatic evaluation and potential circularity

We have provided systematic clarifications to both issues in the revised manuscript and added corresponding analyses in the main text and appendix.
Below we summarize where these additions appear and their core conclusions.

---

### Note · Authors · 2025-12-03

**Comment:**

We sincerely thank the reviewers for the time and effort they dedicated to evaluating our submission, as well as for their constructive and encouraging comments regarding our work on *subjective ability evaluation*, *persona modeling logic*, *behavioral consistency analysis*, and *virtual student agent design*.

The goal of **EduPersona** is to establish a systematic benchmark for subjective ability assessment, aimed at characterizing the capability boundaries of virtual student agents in subjective behavioral dimensions such as emotion perception, help-seeking strategies, cognitive-state expression, and multi-turn interaction stability. The main contributions of the work are three-fold:

* **We define subjective ability boundaries for educational behaviors and construct a multidimensional evaluation framework** covering emotion understanding, help-seeking, interaction style, and cognitive biases;
* **We build a cross-topic, cross-task benchmark for subjective behavioral assessment**, incorporating IRF–BEC structures, persona-expression consistency, multi-turn stability, and structural bias analysis, providing a systematic validation pathway for subjective capabilities in virtual student agents;
* **We introduce a human–AI alignment perspective through human–model consistency evaluation**, quantifying the upper bound of human agreement via expert assessment and analyzing alignment between experts and GPT-4o, thereby validating the stability and absence of significant structural bias in the automatic evaluation module across different model families and persona types.

This work aims to construct a benchmark for subjective ability evaluation, focusing on defining the subjective behavioral capability boundaries of virtual students, rather than developing an applied teaching system or directly validating downstream instructional effectiveness. However, several comments approached the work from the perspective of applied systems or post-training evaluation, raising expectations related to external benchmarks, circularity, or cross-persona comparisons, positions that fall outside the evaluation layer that EduPersona targets.

To address these conceptual differences, we clarified and supplemented the rebuttal and revised manuscript with:

* Systematic presentation of inter-expert agreement and alignment results between GPT-4o and human experts on subjective tasks;
* Additional explanation of acceptance rates and robustness of the automatic evaluator across different model families and ten persona types, clarifying its reliability boundary in subjective evaluation scenarios;
* Clearer definition of persona-label origins, behavioral-consistency measures, and their theoretical grounding, further strengthening the conceptual and operational framing of “subjective ability boundaries.”

At the same time, due to the large-scale anonymity breach that occurred at ICLR this year, the conference implemented special policies to safeguard fairness in the review process, preventing further author–reviewer clarification in later stages. **Within this objective constraint, we recognize that the limited communication space in the current mechanism makes it difficult for a benchmark-oriented work such as this one to have its structural contributions and evidence chain fully presented and fairly evaluated.** Out of respect for scientific rigor and the review process, we have decided to voluntarily withdraw the submission. We will continue to refine the manuscript based on the reviewers’ comments and rebuttal clarifications, and later submit to a venue more suitable for presenting research of this nature.

**Withdrawal Confirmation:**

I have read and agree with the venue's withdrawal policy on behalf of myself and my co-authors.